# Black holes, heavy states, phase shift and anomalous dimensions

**Manuela Kulaxizi, Gim Seng Ng and Andrei Parnachev**

School of Mathematics and Hamilton Mathematics Institute,
Trinity College Dublin, Dublin 2, Ireland

## Abstract

We compute the phase shift of a highly energetic particle traveling in the background of an asymptotically AdS black hole. In the dual CFT, the phase shift is related to a four point function in the Regge limit. The black hole mass is translated to the ratio between the conformal dimension of a heavy operator and the central charge. This ratio serves as a useful expansion parameter; its power measures the number of stress tensors appearing in the intermediate channel. We compute the leading term in the phase shift in a holographic CFT of arbitrary dimensionality using Conformal Regge Theory and observe complete agreement with the gravity result. In a two-dimensional CFT with a large central charge the heavy-heavy-light-light Virasoro vacuum block reproduces the gravity phase shift to all orders in the expansion parameter. We show that the leading order phase shift is related to the anomalous dimensions of certain double trace operators and verify this agreement using known results for the latter. We also perform a separate gravity calculation of these anomalous dimensions to second order in the expansion parameter and compare with the phase shift expansion.


doi:10.21468/SciPostPhys.6.6.065

# 1  Introduction and summary

The AdS/CFT correspondence [1,2,22] is an extremely rich subject. Following the impressive success of the conformal bootstrap [3,31], there was a renewed interest in CFT techniques (see [4–6] for recent reviews). As a result, a number of theoretical instruments, useful for exploring the mechanisms of AdS/CFT in detail were developed. The basic objects on the gravity side of the AdS/CFT correspondence are Witten diagrams [2], which admit a simple decomposition in terms of CFT conformal blocks [63]. In particular, a tree-level Witten diagram with a single graviton exchange gives rise to conformal blocks of spin-two double trace operators, in addition to the stress-tensor conformal block.

Ref. [7] defined a *holographic CFT* as a CFT with a large central charge and a large gap in the spectrum of operator dimensions for operators with spin greater than two. It turns out that considering a certain kinematical limit of four-point functions in such CFTs, (the Regge limit), leads to a set of interesting results. In particular, refs. [56–59] showed that four-point functions in the Regge limit are related to high energy scattering of two particles in AdS. The eikonal approximation to scattering, valid in the Regge limit, gives rise to a phase shift which is proportional to the propagator in the transverse plane ($H_{d-1}$ for the AdS$_{d+1}$ case). As explained in [10, 11], the alternative description of scattering in the Regge limit is provided by considering a highly energetic particle propagating in a shock wave background. The time delay times the lightcone momentum is precisely the eikonal phase shift.

In [8] it was shown that when one of the particles is a graviton, once generic higher derivative terms are added to the Einstein action, there is always a polarization choice which leads to time advance, as opposed to time delay. One can also use Conformal Regge Theory [58] to see how in holographic CFTs the phase shift becomes negative, and unitarity gets violated, unless the three-point couplings of the stress tensor satisfy two linear constraints (which reduce to the "$a = c$" condition in $d = 4$ superconformal field theories). This was done in [9,52,60–62].[1] Note that we now have a definition of the phase shift entirely in terms of a CFT object (the Fourier transform of a four-point function).

---

[1] These strong constraints in holographic CFTs are obtained in the limit of small impact parameter. In the opposite limit of large impact parameter, Hofman-Maldacena [12] and related constraints are recovered [13–21, 32, 33, 41–43, 47].

In this paper we consider a four-point function of scalar operators in holographic CFTs. We take two operators, $\mathcal{O}_H$, to be heavy: their conformal dimension scales with the central charge $\Delta_H \sim C_T$ and the ratio $\mu \sim \Delta_H/C_T$ provides an important parameter. The remaining two operators, $\mathcal{O}_L$, have conformal dimension of order one. We compute the phase shift to leading order in $\mu$ in a $d$−dimensional CFT and show that it is related to the time delay and angular deflection, which an energetic particle experiences when traveling in the background of an asymptotically AdS black hole. The parameter $\mu$ is proportional to the mass of the black hole. In gravity we compute the phase shift to all orders in $\mu$ – for a generic spacetime dimension it remains to be seen whether the CFT result agrees with that. (Terms proportional to $\mu^k$ with $k > 1$ are technically more difficult to compute in the CFT since they require summation over an infinite number of conformal families which schematically correspond to $(T_{\mu\nu})^k$ operators.) However in the $d = 2$ case the CFT result can be obtained to all orders in $\mu$. It is provided by the Virasoro vacuum block, and the result precisely matches the phase shift experienced by a particle traveling in the $AdS_3$ background with a conical deficit.

The rest of the paper is organized as follows. In the next Section we analyze the trajectory of a highly energetic particle, traveling along a null geodesic in the AdS-Schwarzschild background. We compute the time delay and the angular deflection, order by order in the black hole mass $\mu$. In $d = 2$ it is easy to write down the result to all orders; in higher dimensions the answer is more involved, but still manageable.

In Section 3 we use Conformal Regge Theory to compute the phase shift in a holographic CFT to leading order in $\mu$ and observe precise agreement with the gravity result. We show that the double-trace operators (of the type $\mathcal{O}_L \partial_\mu \partial_\nu \mathcal{O}_L$ and $\mathcal{O}_H \partial_\mu \partial_\nu \mathcal{O}_H$) do not contribute to the phase shift.

In Section 4 we use the heavy-heavy-light-light Virasoro vacuum block in a two-dimensional CFT to compute the phase shift to all orders in $\mu$ – the result is a precise match to the gravity calculation.

In Section 5 we show that to leading order in $\mu$, the anomalous dimensions of the double-twist operators in the cross (S-) channel are related to the phase shift. We verify this relation by comparing the $\mathcal{O}(\mu)$ term in the phase shift with the known anomalous dimensions of the $\mathcal{O}_H \partial^\ell \partial^{2n} \mathcal{O}_L$ operators. Such anomalous dimensions are known exactly in $d = 2$ and in the lightcone limit $\ell \gg n \gg 1$ in general $d$. We observe exact agreement. The agreement does not have to extend beyond $\mathcal{O}(\mu)$ and indeed, in $d = 2$ the correspondence between the anomalous dimensions and the phase shift appears to break down at higher orders in $\mu$ (the overall coefficients are not the same).

In Section 6 we compute the anomalous dimensions of the double twist operators $\mathcal{O}_H \partial^\ell \partial^{2n} \mathcal{O}_L$ to order $\mu^2$ in the lightcone limit. This is done by computing the shift in the energies of the corresponding states in the background of the asymptotically AdS black hole. The resulting behavior [eq. (6.42)] has the same scaling as the corresponding term in the phase shift, but a different numerical coefficient.

In Section 7 we discuss our results and some open problems. Appendices contain some technical details needed in the main text.

## 2 Phase shift calculation in gravity

### 2.1 Setting up the problem

Consider an asymptotically AdS black hole in $(d + 1)$-dimension (with AdS radius $R$):

$$ds^2 = -f\,dt^2 + f^{-1}dr^2 + r^2 d\Omega^2, \tag{1}$$

where

$$d\Omega^2 = d\varphi^2 + \sin^2\varphi \; d\Omega_{d-2}^2\,, \tag{2}$$

and

$$f = 1 + \frac{r^2}{R^2} - \frac{\mu}{r^{d-2}} \;\;,\;\; \mu \equiv \left[\frac{d-1}{16\pi}\Omega_{d-1}\right]^{-1}G_N M\,. \tag{3}$$

The Hawking temperature $T_H$ is [45]

$$T_H = \frac{dr_H^2 + (d-2)R^2}{4\pi R^2 r_H}\,, \tag{4}$$

where $r_H$ denotes the position of the horizon:

$$f(r = r_H) = 0\,. \tag{5}$$

The two Killing vectors, $\partial_t$ and $\partial_\varphi$, of 1 allow one to define quantities conserved along the geodesics, *i.e.*, the energy and angular momentum:

$$p^t = \left(1 + \frac{r^2}{R^2} - \frac{\mu}{r^{d-2}}\right)\frac{\partial t}{\partial \lambda}, \qquad p^\varphi = r^2 \frac{\partial \varphi}{\partial \lambda}\,, \tag{6}$$

with $\lambda$ denoting an affine parameter. The equation describing null geodesics becomes

$$\frac{1}{2}\left(\frac{\partial r}{\partial \lambda}\right)^2 + V_{eff}(r) = \frac{1}{2}(p^t)^2\,, \tag{7}$$

where

$$V_{eff}(r) = \frac{(p^\varphi)^2}{2r^2}f(r)\,. \tag{8}$$

A light ray starting from the boundary, traversing the bulk and reemerging on the boundary, will experience both a time delay and a deflection given by

$$\Delta t = 2\int_{r_0}^{\infty}\frac{dr}{f\sqrt{1-\frac{\alpha^2}{r^2}f}}\,,$$

$$\Delta\varphi = 2\alpha\int_{r_0}^{\infty}\frac{dr}{r^2\sqrt{1-\frac{\alpha^2}{r^2}f}}\,. \tag{9}$$

Here, $\alpha = p^\varphi/p^t$ and $r_0$ denotes the turning point of the geodesic, whose existence ensures that the light ray will reach the boundary. It is the minimum point of the trajectory, given by the loci of real and positive $r$ for which $\dot{r} = 0$:

$$1 - \frac{\alpha^2}{r_0^2}f(r_0) = -\left(\frac{\alpha}{r_0}\right)^2\left(-\frac{r_0^2}{b^2} + 1 - \frac{\mu}{r_0^{d-2}}\right) = 0\,. \tag{10}$$

Note that in the second equality we used $b$ to denote

$$b = \left(\frac{(p^t)^2}{(p^\varphi)^2} - \frac{1}{R^2}\right)^{-\frac{1}{2}}\,, \tag{11}$$

which corresponds to the *impact parameter* in pure AdS, as can be easily seen from 10 by setting $\mu = 0$. Clearly $b$ reduces to the familiar flat space expression, $b \approx \frac{p^\varphi}{p^t}$, for large $R$ (or small $p^\varphi/p^t$) whereas it diverges in the limit $p^\varphi/p^t \to R$.

It will be convenient in the following to use the parameterization $b = R \sinh L$, stemming from the standard parametrization of the global AdS metric:

$$ds_{AdS}^2 = R^2 \left( -\cosh^2 L \, dt^2 + dL^2 + \sinh^2 L \, d\Omega^2 \right). \tag{12}$$

Note that 11 implies the following relation between $L$ and $p^\varphi, p^t$:

$$e^{2L} = \frac{p^+}{p^-}, \qquad p^\pm = p^t \pm \frac{p^\varphi}{R}, \tag{13}$$

or, equivalently,

$$\cosh L = \frac{1}{2} \frac{p^+ + p^-}{\sqrt{-p^2}} \quad , \quad p^2 = -p^+ p^-. \tag{14}$$

These relations, between the impact parameter and the momentum of the particle, will be important for the CFT calculations in the following sections.

In this note we are interested in the bulk phase shift. For a particle described by a plane wave, the bulk phase shift is:

$$\delta \equiv -p \cdot (\Delta x) = p^t (\Delta t) - p^\varphi (\Delta \varphi), \tag{15}$$

with $p^{t,\varphi}$ denoting the momenta of the particle traversing the geometry. Combining 15 with 9 yields:

$$\delta(\sqrt{-p^2}, L) = 2 \sqrt{-p^2} \cosh L \int_{r_0}^\infty dr \, \frac{\sqrt{1 - f(r) \frac{R^2}{r^2} \tanh^2 L}}{f(r)}. \tag{16}$$

In pure AdS, the bulk phase shift takes the form:

$$\delta_{AdS} = \pi R \sqrt{-p^2} \, e^{-L}, \tag{17}$$

while $\Delta t = R(\Delta \varphi) = R \pi$: all null geodesics converge at the same point.

The main objective of this section is to compute corrections to the bulk phase shift away from pure AdS, due to the presence of the black hole. We will thus expand and evaluate 15 order by order in $\mu$, in terms of the energy $\sqrt{-p^2}$ and the impact parameter $L$ of the particle.

## 2.2 Small mass expansion of the bulk phase shift

In this section we study the phase shift perturbatively in $\mu$. We will focus on the linear and quadratic terms in the mass and then generalise our results to any order in $\mu$. To appreciate the importance of the small mass expansion, consider for instance $d = 4$ and notice that

$$\frac{r_H^2}{R^2} \sim \frac{\mu}{R^2} \sim \frac{\ell_p^3 M}{R^2} \sim \frac{\Delta_H}{C_T}, \tag{18}$$

where we used $R^3/\ell_p^2 \sim C_T$ and $\Delta_H = M R$. From 18 we deduce that in terms of the dual CFT, the $\mu$-expansion is an expansion in powers of $\Delta_H/C_T$, where $\Delta_H$ corresponds to the conformal dimension of the heavy operator effectively producing a thermal state and $C_T$ is the coefficient of the stress tensor two-point function. Similar arguments hold for general $d$.

To address the small $\mu$-expansion of the bulk phase shift, it is convenient to set $R = 1$ and define a new variable of integration $y = \frac{r_0}{r}$. Next, one would like to eliminate the dependence of the integral on $r_0$ in favour of $\mu$ using 10. It turns out that it is easier to do the opposite, *i.e.*, to eliminate the dependence of the integral on $\mu$ in favour of $r_0$ instead. With a bit of algebra one can show that the bulk phase shift can be expressed as:

$$\delta = 2\sqrt{-p^2}\, b \int_0^1 dy \frac{\sqrt{1-y^2}\sqrt{1-v_0^2 \frac{1-y^d}{1-y^2}}}{(y^2+b^2)\left(1-v_0^2 \frac{y^d+b^2}{y^2+b^2}\right)}, \tag{19}$$

where the natural expansion parameter is now

$$v_0^2 \equiv 1 - \frac{r_0^2}{b^2}. \tag{20}$$

To see this recall that in pure AdS where $\mu = 0$, $r_0 = b$ and thus $v_0$ vanishes as well. To compute the first order term, we take into account that

$$v_0^2 = \sum_{k=1}^{\infty} c_k \mu^k, \tag{21}$$

which follows trivially from 10. The coefficients $c_k$ are computable to any order in $\mu$ and take the form:

$$c_k = \frac{1}{k!} b^{-k(d-2)} \frac{\Gamma\left[k\frac{d}{2}-1\right]}{\Gamma\left[k\frac{d}{2}-k\right]}. \tag{22}$$

The first and second order terms for instance, are:

$$c_1 = b^{-(d-2)}, \qquad c_2 = \frac{d-2}{2} b^{-2(d-2)}. \tag{23}$$

With the help of 21, 23 and 19 the leading order correction reads:

$$\delta_1 = c_1 \left.\frac{\partial \delta}{\partial v_0^2}\right|_{v_0^2=0} = \mu\sqrt{-p^2}\, b^{3-d} \int_0^1 dy \frac{y^d(-2+y^2-b^2)+y^2-b^2+2b^2y^2}{\sqrt{1-y^2}(y^2+b^2)^2}. \tag{24}$$

Notice that certain terms are total derivatives

$$\begin{aligned}
\frac{y^2-b^2+2b^2y^2}{\sqrt{1-y^2}(y^2+b^2)^2} &= -\frac{d}{dy}\frac{y\sqrt{1-y^2}}{(y^2+b^2)}, \\
\frac{y(-2-b^2+y^2)}{\sqrt{1-y^2}(y^2+b^2)^2} &= \frac{d}{dy}\frac{\sqrt{1-y^2}}{(y^2+b^2)},
\end{aligned} \tag{25}$$

allowing us to express 24 as

$$\begin{aligned}
\delta_1 &= \mu\sqrt{-p^2}\, b^{3-d} \left\{ \left(\frac{y\sqrt{1-y^2}}{y^2+b^2} + y^{d-1}\frac{\sqrt{1-y^2}}{y^2+b^2}\right)\Bigg|_{y=0}^{y=1} - (d-1)\int_0^1 dy \frac{y^{d-2}\sqrt{1-y^2}}{y^2+b^2} \right\} = \\
&= \mu\sqrt{-p^2}\, \frac{d-1}{2} B\left[\frac{d-1}{2},\frac{3}{2}\right] b^{1-d} \, {}_2F_1[1,\frac{d-1}{2},\frac{d}{2}+1,-\frac{1}{b^2}].
\end{aligned} \tag{26}$$

Here $B[x,y]$ denotes the Beta function $B[x,y] \equiv \frac{\Gamma(x)\Gamma(y)}{\Gamma(x+y)}$. Using the following identity for hypergeometric functions,

$$_2F_1[a_1, a_2, a_1 - a_2 + 1, w] = (1-w)^{-a_1} \, _2F_1\left[\frac{a_1}{2}, \frac{a_1+1}{2} - a_2, a_1 - a_2 + 1, -\frac{4w}{(1-w)^2}\right], \tag{27}$$

with $w = e^{-2L}$ and $a_1 = d - 1$, $a_2 = \frac{d}{2} - 1$, and setting $b = \sinh L$, leads to the more familiar form [56, 57] [2]

$$\delta_1 = \mu \sqrt{-p^2} \frac{d-1}{2} B\left[\frac{d-1}{2}, \frac{3}{2}\right] 2^{d-1} e^{-(d-1)L} \, _2F_1\left[d-1, \frac{d}{2} - 1, \frac{d}{2} + 1, e^{-2L}\right]$$

$$\implies \qquad \delta_1 = \mu(d-1)\frac{\pi^{\frac{d}{2}}}{\Gamma[\frac{d}{2}]} \sqrt{-p^2} \, \Pi_{d-1;d-1}(L), \tag{29}$$

where $\Pi_{\Delta-1;d-1}$ denotes the Euclidean hyperbolic space $H_{d-1}$ propagator for a massive particle of mass-square equal to $(\Delta - 1)^2$, defined as

$$\Pi_{\Delta-1;d-1}(x) = \frac{\pi^{1-\frac{d}{2}}\Gamma(\Delta-1)}{2\Gamma(\Delta - \frac{d-2}{2})} \, e^{-(\Delta-1)x} \, _2F_1(\frac{d}{2} - 1, \Delta - 1, \Delta - \frac{d}{2} + 1, e^{-2x}). \tag{30}$$

Moving on to the second order term in $\mu$, we write:

$$\delta_2 = \mu^2 \frac{1}{2} \sqrt{-p^2} \left(2c_2 \left.\frac{\partial \delta}{\partial v_0^2}\right|_{v_0^2=0} + c_1^2 \left.\frac{\partial^2 \delta}{\partial (v_0^2)^2}\right|_{v_0^2=0}\right), \tag{31}$$

and evaluate the derivatives using 19. The resulting integrand is given by a rather lengthy expression, but one can still evaluate the integral by splitting it into two parts: a total derivative term and another one which coincides with the representation of a certain hypergeometric function (the interested reader may consult Appendix A for details). The final result for the quadratic term in $\mu$ can be expressed as follows:

$$\delta_2 = \mu^2 \frac{(2d-3)(2d-1)}{4} \frac{\pi^{d-1}}{\Gamma[d-1]} \sqrt{-p^2} \, \Pi_{2d-3,2d-3}(L). \tag{32}$$

It is instructive to notice that the quadratic result is not proportional to the hyperbolic space propagator in $H_{d-1}$ but rather in $H_{2d-3}$, as if the dimensionality of the space is $2(d-1)$ instead of $d$. It is possible to evaluate the bulk phase shift to arbitrary order in the $\mu$-expansion. Having checked several higher order terms in the $\mu$ expansion, we deduce that the bulk phase shift can be written as:

$$\delta(\sqrt{-p^2}, L) = \sum_{k=0}^{\infty} \delta_k(\sqrt{-p^2}, L) =$$

$$= \sum_{k=0}^{\infty} \frac{\mu^k}{k!} \frac{2\Gamma\left[\frac{dk+1}{2}\right]}{\Gamma\left[\frac{k(d-2)+1}{2}\right]} \frac{\pi^{\frac{k(d-2)+2}{2}}}{\Gamma\left[\frac{k(d-2)+2}{2}\right]} \sqrt{-p^2} \, \Pi_{k(d-2)+1,k(d-2)+1}(L). \tag{33}$$

---

[2]To arrive at the last line we used the identity:

$$\Gamma\left[x - \frac{1}{2}\right] = 2^{2-2x} \pi^{\frac{1}{2}} \frac{\Gamma[2x-1]}{\Gamma[x]} \tag{28}$$

In other words the $k$-th order term in the bulk phase shift is proportional to the propagator of a massive particle of mass-square $m^2 = k(d-1) - (k-1)$ in a hyperbolic space of the same dimensionality, $k(d-1) - (k-1)$. Furthermore, in $d = 4$, a closed form solution for the phase shift can be found in terms of either elliptic integrals or Appell $F_1$ functions. Precise expressions can be found in Appendix B.

## 2.3 Exact solution in $d = 2$

The case of $d = 2$ is special, since the last term in $f(r)$ in 3 is $r-$independent. In fact, for $\mu < 1$ there is no horizon and the geometry is just AdS$_3$ space with a conical deficit. The metric 1 can be written as the AdS$_3$ metric (we set $R = 1$)

$$ds^2 = -\left(1 + \tilde{r}^2\right)d\tilde{t}^2 + \left(1 + \tilde{r}^2\right)^{-1}d\tilde{r}^2 + \tilde{r}^2 d\tilde{\varphi}^2, \tag{34}$$

where

$$\tilde{\varphi} = \sqrt{1-\mu}\,\varphi, \qquad \tilde{t} = \sqrt{1-\mu}\,t, \qquad \tilde{r} = \frac{r}{\sqrt{1-\mu}}. \tag{35}$$

A null geodesic which starts from the boundary at $\tilde{\varphi} = 0, \tilde{t} = 0$ arrives back to the boundary at $\tilde{\varphi} = \pi$ and $\tilde{t} = \pi$. We can now translate this to the original coordinates $\varphi, t$ to obtain the time delay and the angular deflection of the particle's trajectory:

$$\Delta t = \left(\frac{1}{\sqrt{1-\mu}} - 1\right)\pi, \qquad \Delta\varphi = \left(\frac{1}{\sqrt{1-\mu}} - 1\right)\pi, \tag{36}$$

which gives

$$\Delta x^+ = 2\pi\left(\frac{1}{\sqrt{1-\mu}} - 1\right), \qquad \Delta x^- = 0. \tag{37}$$

The phase shift is given by

$$\delta = \frac{1}{2}p^-\Delta x^+ = \pi\sqrt{-p^2}\,e^{-L}\left(\frac{1}{\sqrt{1-\mu}} - 1\right), \tag{38}$$

where we used 13. It is instructive to expand 38 in powers of $\mu$:

$$\delta = \pi\sqrt{-p^2}\,e^{-L}\left(\frac{\mu}{2} + \frac{3\mu^2}{8} + \frac{5\mu^3}{16} + \dots\right). \tag{39}$$

It agrees with eq. 33 upon the substitution of $d = 2$ in the latter. Note that the $d = 2$ case is special, as for $\mu < 1$ the geometry is described by a defect as opposed to a black hole. In the BTZ case ($\mu > 1$), the null geodesics discussed above do not return to the boundary. This is related to the divergence of 38 as $\mu \to 1$. The meaning of 38 when analytically continued to $\mu > 1$ deserves further exploration.

# 3 CFT calculation of the phase shift

## 3.1 Kinematics

The main object of study is the four-point function on the cylinder parameterized by time $\tau$ and a point on the $d-1$-dimensional unit sphere, $(\hat{n})$:

$$\langle \mathcal{O}_H^{cyl}(\tau_4, \hat{n}_4)\mathcal{O}_L^{cyl}(\tau_3, \hat{n}_3)\mathcal{O}_L^{cyl}(\tau_2, \hat{n}_2)\mathcal{O}_H^{cyl}(\tau_1, \hat{n}_1)\rangle, \tag{40}$$

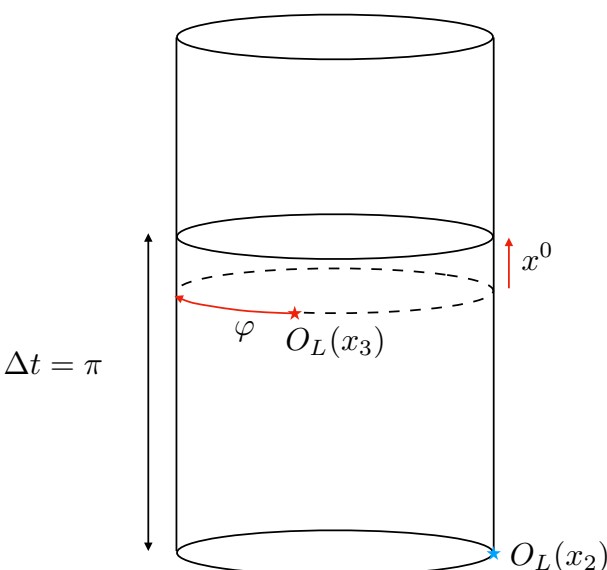

Figure 1: Positions of the light operators on the cylinder. The states at both ends of the cylinder are created by the heavy operators $\mathcal{O}_H$.

where $\mathcal{O}_H^{cyl}$ is a heavy operator whose dimension $\Delta_H \sim C_T$ and $\mathcal{O}_L^{cyl}$ is a light operator with dimension $\Delta_L$ which scales as $\Delta_L \sim O(1)$.[3] The heavy operators are inserted at $\tau_1 = -\infty, \tau_4 = \infty$; via the operator-state correspondence, they correspond to heavy states.

Note that with the superscripts 'cyl', these are operators on the cylinders. The map between the operators on the cylinder and on the plane is

$$\mathcal{O}^{plane}(x) = e^{-\tau \Delta_{\mathcal{O}}} \mathcal{O}^{cyl}(\tau, \hat{n}) \quad , \quad x^2 = e^{2\tau}. \tag{41}$$

We will keep these superscripts in some instances, but in order to avoid cluttering of notations we will drop them whenever their meanings are clear or insignificant in the context.

The other two operators (i.e. the light operators) are inserted close to two reference points $P_2$ and $P_3$ on the cylinder. The reference points $P_2$ and $P_3$ differ by $\delta t = \pi$ in the Lorentzian time (related to the Euclidean time by the usual Wick rotation, $t = i\tau$). In addition, the sphere coordinates of $P_2$ and $P_3$ are diametrically opposite: $\hat{n}(P_2) = -\hat{n}(P_3)$. Note that $P_2$ and $P_3$ define Poincare patches centered over them. The coordinates of insertions on the cylinder $x = (x^0, \hat{n})$ relative to $P_2$ and $P_3$ are precisely the $x$-coordinates we need to Fourier transform over. We will use the translational symmetry in $\tau$ together with the rotational symmetry of $S^{d-1}$ to fix the position of $\mathcal{O}(\tau_2, \hat{n}_2) = \mathcal{O}(P_2)$. The position of $\mathcal{O}(\tau_3, \hat{n}_3)$ is parameterized by[4]

$$\tau_3 = \tau_2 - i\delta t, \qquad \delta t = \pi + x^0 \quad ; \quad x^0 \geq 0 \tag{42}$$

and $\hat{n}_3 = \hat{n}$. The kinematics are summarized in Fig. 1. The cylinder correlator 40 can now be transformed to the plane $R^d$ via the usual map from the euclidean time on the cylinder to the radial polar coordinate $r = e^\tau$. We can now use this to go from $\tau_1 = -\infty, \tau_4 = +\infty$ to

---

[3]For a recent discussion of four-point function in a similar context see e.g. [53].

[4]Note that $x^0 \geq 0$ implies that $\mathcal{O}(\tau_3, \hat{n}_3)$ is future-time-like with respect to $\mathcal{O}(P_2)$.

$x_1 = 0, x_4 = \infty$ and write

$$\langle \mathcal{O}_H^{cyl} | \mathcal{O}_L^{cyl} \mathcal{O}_L^{cyl} | \mathcal{O}_H^{cyl} \rangle = (r_2 r_3)^{\Delta_L} \lim_{x_4 \to \infty} (x_4^2)^{\Delta_H} \langle \mathcal{O}_H^{plane}(x_4) \mathcal{O}_L^{plane}(x_3) \mathcal{O}_L^{plane}(x_2) \mathcal{O}_H^{plane}(0) \rangle, \tag{43}$$

where the factor $(r_2 r_3)^{\Delta_L}$ appears due to the conformal transformation from the cylinder to $R^d$. This can be further written as

$$A(x) \equiv \langle \mathcal{O}_H^{cyl} | \mathcal{O}_L^{cyl}(x_3) \mathcal{O}_L^{cyl}(x_2) | \mathcal{O}_H^{cyl} \rangle = (r_2 r_3)^{\Delta_L} \times \frac{\mathcal{A}(u,v)}{x_{32}^{2\Delta_L}}, \tag{44}$$

where we have defined the partial amplitude $\mathcal{A}(u,v)$ which only depends on cross-ratios and can be expanded in conformal blocks. In our conventions and setup, the cross-ratios are

$$z\bar{z} = u = \frac{x_2^2}{x_3^2} = e^{2(\tau_2 - \tau_3)} = e^{2i\delta t} = e^{2ix^0} \tag{45}$$

and

$$(1-z)(1-\bar{z}) = v = \frac{x_{32}^2}{x_3^2} = 1 + e^{2(\tau_2 - \tau_3)} - 2e^{\tau_2 - \tau_3} \hat{n}_3 \cdot \hat{n}_2 = 1 + e^{2i\delta t} + 2e^{i\delta t} \cos\varphi, \tag{46}$$

where $\varphi$ is the angle between $\hat{n}(P_3) = -\hat{n}_2$ and $\hat{n}$. Substituting 42 this becomes

$$(1-z)(1-\bar{z}) = 1 + e^{2ix^0} - 2e^{ix^0} \cos\varphi. \tag{47}$$

We can solve 45 and 47 to obtain

$$z = e^{ix^+}, \qquad \bar{z} = e^{ix^-}, \tag{48}$$

where $x^+ = x^0 + \varphi$, $x^- = x^0 - \varphi$. We would like to study the Lorentzian correlator in the limit where $x^\pm$ are small – this is the Regge limit discused in [56,57].

The configuration in question is reached by starting from the correlator where $\mathcal{O}_3$ and $\mathcal{O}_2$ are inserted close to each other on a spatial circle and at the same time. This corresponds to $x^+ \approx -2\pi$. To reach the configuration where $\mathcal{O}_3$ is inserted close to $P_3$, we need to shift $x^+ \to x^+ + 2\pi$ which corresponds to $z \to e^{2\pi i} z$. Note that in 48 we could have had the opposite assignments of $z, \bar{z}$ – there is a complete symmetry of the correlator, before the analytic continuation. Now we can define the phase shift $\delta(p)$ via the Fourier transform of the correlator $A(x)$ in 44 in the Regge limit where $x^\pm$ are small:[5]

$$\mathcal{B}(p) = \int d^d x A(x) e^{-ipx} \simeq \int d^d x e^{-ipx} \frac{\mathcal{A}(x)}{(-x^2 - i\epsilon x^0)^{\Delta_L}} \equiv \mathcal{B}_0(p) e^{i\delta}, \tag{49}$$

where $\mathcal{B}_0(p)$ denotes the Fourier transform of the disconnected correlator (the contribution from the identity operator),

$$\mathcal{B}_0(p) \equiv \int d^d x e^{-ipx} \frac{1}{(-x^2 - i\epsilon x^0)^{\Delta_L}} = \theta(p^0)\theta(-p^2) e^{i\pi\Delta_L} C(\Delta_L)(-p^2)^{\Delta_L - \frac{d}{2}}, \tag{50}$$

with

$$C(\Delta) \equiv \frac{2^{d+1-2\Delta} \pi^{1+\frac{d}{2}}}{\Gamma(\Delta_L) \Gamma\left(\Delta_L - \frac{d}{2} + 1\right)}. \tag{51}$$

The $i\epsilon$-prescription is inherited from the ordering of the operators, which translates into sending $\delta t = t_3 - t_2 \to \delta t - i\epsilon/2$ with $\epsilon > 0$. This then becomes $x^2 \to x^2 + i\epsilon x^0$. Finally, note that in writing 49 we have assumed that the phase shift exponentiates.

---

[5]Note that our $x$ here differs by a minus sign from that in [52].

### 3.2 Phase shift to $\mathcal{O}(\mu)$ : conformal Regge theory

To compute the phase shift to leading order in $\mu$, it is convenient to parametrise $(z, \bar{z})$ in terms of the variables $(\sigma, \rho)$ defined via:

$$1 - z = \sigma e^\rho, \qquad 1 - \bar{z} = \sigma e^{-\rho}. \tag{52}$$

Expanding 48 to first order in $x^\pm$ leads to

$$\sigma = e^{-\frac{i\pi}{2}}\sqrt{-x^2}, \qquad \cosh\rho = \frac{1}{2}\frac{x^+ + x^-}{\sqrt{-x^2}}. \tag{53}$$

Note that $\sigma$ here is purely imaginary, while $\rho$ is real. It is convenient to set $x^\nu = \sqrt{-x^2}e^\nu$ with $e^2 = -1$, and use this to express $\cosh\rho$ in 53 as follows:

$$\cosh\rho = -e \cdot \bar{e}. \tag{54}$$

Here $\bar{e}$ denotes a fixed vector with all components set to zero except for $\bar{e}^0 = 1$. The leading connected contribution to the correlator can be computed using conformal Regge theory [58].

In the limit $\sigma \to 0$, assuming that the leading Regge contribution comes from an operator of dimension $\Delta$ and spin $j$

$$\mathcal{A} = 1 - 2\pi i \int_{-\infty}^{+\infty} d\nu\, r[\Delta(j(\nu)), j(\nu)]\alpha(\nu)\, \sigma^{1-j(\nu)}\Omega_{i\nu}(\rho) + \dots, \tag{55}$$

where

$$\alpha(\nu) \equiv -\frac{\pi^{\frac{d}{2}-1}4^{j(\nu)+1}e^{-i\pi j(\nu)/2}}{2\sin\left(\frac{\pi j(\nu)}{2}\right)}\beta(\nu)\Gamma\left(\frac{2\Delta_L + j(\nu) - \frac{d}{2} + i\nu}{2}\right)\Gamma\left(\frac{2\Delta_L + j(\nu) - \frac{d}{2} - i\nu}{2}\right),$$
$$\beta \equiv \frac{\pi}{4\nu}j'(\nu),$$

$$\tag{56}$$

and

$$\Omega_{i\nu}(\rho) = \frac{i\nu}{2\pi}\left(\Pi_{i\nu+\frac{d}{2}-1} - \Pi_{-i\nu+\frac{d}{2}-1}\right). \tag{57}$$

Here $\Pi_{i\nu+\frac{d}{2}-1} \equiv \Pi_{i\nu+\frac{d}{2}-1,d-1}$ is the propagator in the Euclidean hyperbolic space $H_{d-1}$ defined in 30. Finally, $r[\Delta(j(\nu)), j(\nu)]$ denotes the analytic continuation in spin and conformal dimension of

$$r[\Delta, J] \equiv \lambda_{\mathcal{O}_L\mathcal{O}_L\mathcal{O}}\lambda_{\mathcal{O}_H\mathcal{O}_H\mathcal{O}}\widetilde{K}_{\Delta,J}(\Delta_L), \tag{58}$$

with

$$\widetilde{K}_{\Delta,J} \equiv \frac{\Gamma(\Delta + J)\Gamma(\Delta - \frac{d}{2} + 1)(\Delta - 1)_J}{4^{J-1}\Gamma(\frac{\Delta+J}{2})^4\Gamma(\frac{2\Delta_L-\Delta+J}{2})\Gamma(\frac{2\Delta_L+\Delta+J-d}{2})} \tag{59}$$

and $\lambda_{\mathcal{O}_L\mathcal{O}_L\mathcal{O}}, \lambda_{\mathcal{O}_H\mathcal{O}_H\mathcal{O}}$ the respective OPE coefficients.

Here we are interested in holographic CFTs, *i.e.* large $C_T$ CFTs, with a large gap $\Delta_{gap}$ in the spectrum of operators. In this case, $j(\nu)$ can be approximated by (see e.g. [52])

$$j(\nu) = 2 - 2\frac{\nu^2 + \frac{d^2}{4}}{\Delta_{gap}^2} + \dots. \tag{60}$$

The integral in 55 can then be computed by closing the contour in the lower half plane and picking up the poles of 56, which correspond to the exchange of the stress tensor operator ($\nu = -id/2$), and the double trace operators composed out of $\mathcal{O}_L$ – schematically denoted by $\mathcal{O}_L \partial_{\mu_1} \dots \partial_{\mu_\ell} \partial^{2n} \mathcal{O}_L$. Comparing eqs. 55, 56 and 59 with the analogous expressions defining $\alpha(\nu)$ in [52] where the four-point function of two pairs of light operators was considered, one notices the absence of factors with poles at the conformal dimensions of the double-trace operators $\mathcal{O}_H \partial_{\mu_1} \dots \partial_{\mu_\ell} \partial^{2n} \mathcal{O}_H$. This is a direct consequence of the limit $\Delta_H \sim C_T \gg n, \ell, \Delta_L$ that we are interested in. In this limit, terms involving $\Delta_H$ cancel out and the poles coming from the double-trace operators built out of $\mathcal{O}_H$ disappear. Evaluating the integral, yields the half-geodesic graviton exchange Witten diagram [63] where the geodesic sits in the center of AdS and corresponds to the heavy state. It should be emphasized that this description is only valid when considering the $\mathcal{O}(\mu)$ contribution to the correlator; higher orders in $\mu$ correspond to exchanges of multiple gravitons (for details see the discussion in section 4). We will now perform the Fourier transform with respect to $x$ to compute the phase shift $\delta(p)$ to $\mathcal{O}(\mu)$. According to the previous section, we are interested in the Fourier transform:

$$\int d^d x\, e^{-ipx} \frac{\mathcal{A}(x)}{(-x^2 - i\epsilon x^0)^{\Delta_L}} = B_0(p)\left[1 + i\delta_1 + \mathcal{O}(\mu^2)\right],\tag{61}$$

with $\mathcal{A}(x)$ given to leading order in the Regge limit by 55. The Fourier transform can be computed in a manner similar to the one in e.g. [52].

We will first derive the following identity:

$$\frac{2^{1-a} e^{i\pi a/2}}{\pi^{(d-2)/2}} \int_{M^+} d^d p \frac{e^{ipx}}{(-p^2)^{\frac{d-a}{2}}} \Omega_{i\nu}(\omega \cdot \bar{e}) = \frac{\Gamma\left(\frac{a - \frac{d-2}{2} + i\nu}{2}\right) \Gamma\left(\frac{a - \frac{d-2}{2} - i\nu}{2}\right)}{(-x^2)^{\frac{a}{2}}} \Omega_{i\nu}(e \cdot \bar{e}),\tag{62}$$

where $M^+$ denotes the upper Milne wedge defined by $\left(p^2 < 0, p^0 > 0\right)$. We start by writing

$$e^{i\pi a/2} \int_{M^+} d^d p \frac{e^{ipx}}{(-p^2)^{\frac{d-a}{2}}} \Omega_{i\nu}(\omega \cdot \bar{e}) = \int_{H_{d-1}} d\omega \frac{\Gamma(a)}{(-\omega \cdot x)^a} \Omega_{i\nu}(\omega . \bar{e}),\tag{63}$$

and then use

$$\frac{2^{1-a}}{\pi^{\frac{d-2}{2}}} \frac{1}{(-e . \omega)^a} = \int_{-\infty}^{\infty} d\nu' \frac{\Gamma\left(\frac{a - \frac{d-2}{2} + i\nu}{2}\right) \Gamma\left(\frac{a - \frac{d-2}{2} - i\nu}{2}\right)}{\Gamma(a)} \Omega_{i\nu'}(e . \omega),\tag{64}$$

to express 63 as

$$\frac{2^{1-a} e^{i\pi a/2}}{\pi^{(d-2)/2}} \int_{M^+} d^d p \frac{e^{ipx}}{(-p^2)^{\frac{d-a}{2}}} \Omega_{i\nu}(\omega \cdot \bar{e}) =$$

$$= \int_{-\infty}^{\infty} d\nu' \frac{\Gamma\left(\frac{a - \frac{d-2}{2} + i\nu}{2}\right) \Gamma\left(\frac{a - \frac{d-2}{2} - i\nu}{2}\right)}{(-x^2)^{\frac{a}{2}}} \int_{H_{d-1}} d\omega\, \Omega_{i\nu}(\omega \cdot \bar{e}) \Omega_{i\nu'}(\omega \cdot e).\tag{65}$$

Using the hyperbolic space identity:

$$\int_{H_{d-1}} d\omega\, \Omega_{i\nu}(\omega \cdot \bar{e}) \Omega_{i\nu'}(\omega \cdot e) = \frac{1}{2}\left(\delta(\nu - \nu') + \delta(\nu + \nu')\right)\tag{66}$$

we arrive at 62[6].

---

[6]62 is the analog of eq. (3.25) in [52], first derived in [58, 59]. The Fourier transform there is taken over the positions of two pairs of operators, while here the positions of the pair of heavy operators are fixed at $\pm\infty$ and are not integrated over.

We now use the identity 62 with $a = 2\Delta_L + j(\nu) - 1$, combined with 55, 56, 60 and 62, to compute the Fourier transform 61 and read off the $\mathcal{O}(\mu)$ term in the phase shift. We find that

$$\delta_1 = -\frac{\pi^{d+1} 2^{4-2\Delta_L+d}}{C(\Delta_L)} \int_{-\infty}^{+\infty} d\nu \, \tilde{r}[\Delta(j(\nu)), j(\nu)] \frac{2^{j(\nu)} e^{i\pi j(\nu)/2} \beta(\nu)}{\sin\left(\frac{\pi j(\nu)}{2}\right)} (-p^2)^{\frac{j(\nu)-1}{2}} \Omega_{i\nu}(\omega \cdot \bar{e}), \quad (67)$$

with

$$\omega \cdot \bar{e} = -\frac{p^+ + p^-}{2\sqrt{-p^2}} = -\cosh L. \quad (68)$$

The integral in 67 can be computed by closing the contour in the lower half-plane and picking up the contribution from the stress-tensor pole. The poles corresponding to double trace operators disappeared after the Fourier transform. The result is

$$\delta_1 = \frac{\lambda_{\mathcal{O}_L \mathcal{O}_L T} \lambda_{\mathcal{O}_H \mathcal{O}_H T}}{\Delta_L} \times \left[\frac{8(d-1)d\pi^{\frac{d}{2}} \Gamma(d+2)}{\Gamma\left(\frac{d}{2}+1\right)^3}\right] \times \sqrt{-p^2} \times \Pi_{d-1;d-1}(L). \quad (69)$$

Note however that

$$\lambda_{\mathcal{O}_L \mathcal{O}_L \mathcal{O}} \lambda_{\mathcal{O}_H \mathcal{O}_H \mathcal{O}} \Delta_L^{-1} = \left(\frac{d}{d-1}\right)^2 \frac{\Delta_H}{C_T} = \frac{2d^2 \pi^{1-\frac{d}{2}} \Gamma\left(\frac{d}{2}\right)^3}{(d-1)\Gamma(d+2)} \frac{G_N \Delta_H}{R^{d-1}} = \frac{\mu}{R^{d-2}} \frac{\Gamma\left(\frac{d}{2}+1\right)^2}{\Gamma(d+2)}, \quad (70)$$

where we used the $\text{AdS}_{d+1}/\text{CFT}_d$ dictionary

$$C_T = \frac{\pi^{\frac{d}{2}-1} \Gamma(d+2)}{2(d-1)\Gamma\left(\frac{d}{2}\right)^3} \frac{R^{d-1}}{G_N} \quad (71)$$

and the relation [45]

$$\mu \equiv \left[\frac{d-1}{16\pi} \Omega_{d-1}\right]^{-1} G_N M = \frac{8\pi^{1-\frac{d}{2}} \Gamma\left(\frac{d}{2}\right)}{d-1} \frac{G_N \Delta_H}{R}. \quad (72)$$

Substituting into 70 leads to

$$\delta_1 = \mu \left[\frac{(d-1)\pi^{d/2}}{\Gamma\left(\frac{d}{2}\right)}\right] \times \sqrt{-p^2} \times \Pi_{d-1;d-1}(L), \quad (73)$$

where we set the AdS radius $R$ to unity. The final answer for 69 is exactly the same as 29. For later comparison, it will be useful to determine the behavior of $\delta_1$ in the lightcone limit, which in the variables defined above is given by $L \gg 1$ . It is easy to see that:

$$\delta_1 \approx \mu \left[\frac{\pi \Gamma(d)}{\Gamma\left(\frac{d}{2}+1\right)\Gamma\left(\frac{d}{2}\right)}\right] \sqrt{-p^2} e^{-(d-1)L}. \quad (74)$$

## 4 The case of CFT$_2$

We will start by using the exact result for the heavy-heavy-light-light Virasoro vacuum block to obtain the time delay to all orders in $\mu$. We will then proceed to analyze the expansion in powers of $\mu$. This will be useful for understanding the higher order terms in higher dimensions.

## 4.1 Phase shift to all orders in $\mu$

In the $d = 2$ case we have much better control.

The solution to the cross-ratios is given by 48, where $x^0$ and $\varphi$ are coordinates on a flat two-dimensional cylinder. In $d = 2$ we can make use of the Virasoro heavy-heavy-light-light vacuum block [64–66], which incorporates contributions from an infinite number of quasi-primaries. The result for the correlator is a product of holomorphic and anti-holomorphic parts:

$$A(x) = \langle \mathcal{O}_H | \mathcal{O}(x_3) \mathcal{O}(x_2) | \mathcal{O}_H \rangle \simeq e^{\Delta f(z)} e^{\Delta f(\bar{z})}, \tag{75}$$

where $z, \bar{z}$ are related to the positions of the operators by 48, as before. Note that the correlator factorizes and exponentiates. We have denoted $\mathcal{O} \equiv \mathcal{O}_L$ and $\Delta \equiv \Delta_L$ to simplify the notations. It will be convenient for us to write the function $f(z)$ as [64, 66]

$$f(z) = -\frac{1}{2} \log z - \log \left( -2 \sinh \left[ \frac{\bar{\alpha}}{2} \log z \right] \right) + \log \bar{\alpha}, \tag{76}$$

where $\bar{\alpha} = \sqrt{1 - \mu}$. Hence the Euclidean correlator, up to an unimportant constant, is

$$\langle \mathcal{O}_H | \mathcal{O}(x_3) \mathcal{O}(x_2) | \mathcal{O}_H \rangle \simeq \frac{1}{\left( \sinh \left[ \frac{\bar{\alpha}}{2} \log z \right] \right)^\Delta \left( \sinh \left[ \frac{\bar{\alpha}}{2} \log \bar{z} \right] \right)^\Delta}, \tag{77}$$

where a factor of $z^{-\Delta/2}$ has been taken care of by the conformal factor which we earned transforming from the plane to the cylinder.

The correlator contains an infinite number of poles at

$$\bar{\alpha} \log z = 2\pi n. \tag{78}$$

These simply correspond to null particles propagating in the bulk of $AdS_3$ with a conical defect (the corresponding anti-holomorphic part describes null particles propagating in the opposite direction on the spatial circle). The $x^\pm$ coordinates defined in the previous section measure the distance from the spacetime point $P_3$; this involves the analytic continuation $z \to e^{2\pi i} z$, which yields with the help of 48 (again, up to an unimportant numerical constant)

$$\frac{1}{\left( \sinh \left[ \frac{\bar{\alpha}}{2} \log z \right] \right)^\Delta} \to \frac{1}{\left( \sinh \left[ \pi i \bar{\alpha} + \frac{\bar{\alpha}}{2} \log z \right] \right)^\Delta} = \frac{1}{\left( \sin \left[ \bar{\alpha} \pi + \frac{\bar{\alpha}}{2} x^+ \right] \right)^\Delta}. \tag{79}$$

This implies that the correlator has a pole at the value of $x^+ = x_*^+$ where the sine vanishes, i.e., where its argument is equal to $\pi n$ (with integer $n$).

Note that we must pick the $n = 1$ solution to recover $x_*^+ = 0$ in the $\bar{\alpha} \to 1$ limit, hence

$$\pi \bar{\alpha} + \frac{\bar{\alpha}}{2} x_*^+ = \pi. \tag{80}$$

In other words, we reproduce the expected result: the amplitude has a pole at

$$x_*^+ = 2\pi \left( \frac{1}{\sqrt{1 - \mu}} - 1 \right), \tag{81}$$

and $x_*^- = 0$. This is of course the expected time delay combined with the angular shift of the null geodesic in the conical defect background 37.

In other words, we successfully reproduced the gravity result for the time delay and angular deflection. The phase shift is obtained by multiplying the expression in 81 by $p^-$. This is consistent with

$$\int d^2x \, e^{\frac{1}{2}ip^-x^+} \, e^{\frac{1}{2}ip^+x^-} \, \langle \mathcal{O}_H | \mathcal{O}(x_3) \mathcal{O}(x_2) | \mathcal{O}_H \rangle \approx \mathcal{B}_0(p) \, e^{i\delta} \,, \tag{82}$$

where $\delta$ is given precisely by 38 and $\mathcal{B}_0(p)$,

$$\mathcal{B}_0(p) = C(\Delta_L)\theta(p^-)\theta(p^+)e^{i\pi\Delta_L}(p^+p^-)^{\Delta_L-1} \,, \tag{83}$$

represents the contribution from the disconnected piece. The approximation in 82 is valid in the limit $p^2 \gg 1$; the integral simply picks up the pole given by 81.

The other poles of the correlator 79 correspond to null geodesics in the bulk which bounce from the boundary into the bulk, and reemerge later. We are not interested in them – among other things, the phase shift associated with these poles is larger than the one which corresponds to the geodesic with no bounce ($n = 1$ solution discussed above). Besides, the $n = 1$ solution is the only one which is perturbative in $\mu$ – it is this perturbative expansion we aim to eventually reproduce in the higher dimensional setting.

## 4.2 Expansion of the correlator in powers of $\mu$

As explained in the previous section, the correlator of heavy-heavy-light-light operators is a product of a holomorphic and an anti-holomorphic part, each of which exponentiates, *i.e.* can be written as $e^{\Delta f(z)}e^{\bar{\Delta} f(\bar{z})}$, with $f(z)$ defined in 76. The function $f(z)$ has a simple expansion in terms of $\mu$,

$$f(z) = f_1(z) + f_2(z) + \dots \,, \tag{84}$$

where $f_k(z)$ contains a factor of $\mu^k$. The precise expansion is given by (here $w = 1 - z$):

$$\begin{aligned}
f(w) = &-\ln w + \mu\left[ -\frac{1}{2} + \frac{4(w-2)\ln(1-w)}{w} \right] \\
&+ \mu^2\left[ -\frac{(w-1)\ln^2(1-w)}{8w^2} + \frac{(w-2)\ln(1-w)}{16w} - \frac{1}{4} \right] \\
&+ \mu^3\left[ -\frac{(w-1)(w-2)\ln^3(1-w)}{48w^3} - \frac{(w-1)\ln^2(1-w)}{16w^2} + \frac{(w-2)\ln(1-w)}{32w} - \frac{1}{6} \right] \\
&+ \mu^4\left[ -\frac{(w-1)((w-6)w+6)\log^4(1-w)}{384w^4} - \frac{(w-2)(w-1)\log^3(1-w)}{64w^3} \right. \\
&\qquad \left. -\frac{5(w-1)\log^2(1-w)}{128w^2} + \frac{5(w-2)\log(1-w)}{256w} - \frac{1}{8} \right] \\
&+ \cdots .
\end{aligned} \tag{85}$$

Interestingly, each term in the $\mu$ expansion is a sum of all the possible terms made out of products of holomorphic, global blocks satisfying the following requirements[7]:

- the leading behavior for small $w$ of each product should be equal to that of the stress-tensor block to the $k$-th power.

- the sum of the dimensions/spins of the blocks in each product is equal to $(2k)$, where $k$ indicates the $k$-th term in the $\mu$-expansion.

---

[7]This is actually the generalization of an observation which appeared in Appendix D2 of [72].

To be explicit, the order $\mu, \mu^2, \mu^3, \mu^4$-terms in brackets in 85 can be written as:

$$
\begin{aligned}
\mathcal{O}(\mu): &\ (w^2\ {}_2F_1[2,2,4,w]) \\[4pt]
\mathcal{O}(\mu^2): &\ -\left(w^2\ {}_2F_1[2,2,4,w]\right)^2 + \frac{6}{5}(w^3\ {}_2F_1[3,3,6,w])(w\ {}_2F_1[1,1,2,w]) \\[4pt]
\mathcal{O}(\mu^3): &\ \frac{4}{3}\left(w^2\ {}_2F_1[2,2,4,w]\right)^3 - \frac{14}{5}(w^2\ {}_2F_1[2,2,4,w])(w^3\ {}_2F_1[3,3,6,w])(w\ {}_2F_1[1,1,2,w])+ \\[4pt]
&\ +\frac{2\times 27}{35}(w^4\ {}_2F_1[4,4,8,w])(w\ {}_2F_1[1,1,2,w])^2 \\[4pt]
\mathcal{O}(\mu^4): &\ -2\left(w^2\ {}_2F_1[2,2,4,w]\right)^4 \\[4pt]
&\ +\frac{59}{10}(w^2\ {}_2F_1[2,2,4,w])^2(w^3\ {}_2F_1[3,3,6,w])(w\ {}_2F_1[1,1,2,w])- \\[4pt]
&\ -\frac{297}{70}(w^2\ {}_2F_1[2,2,4,w])(w^4\ {}_2F_1[4,4,8,w])(w\ {}_2F_1[1,1,2,w])^2- \\[4pt]
&\ -\frac{42}{25}(w^3\ {}_2F_1[3,3,6,w])^2(w\ {}_2F_1[1,1,2,w])^2 \\[4pt]
&\ +\frac{72}{5}(w^5\ {}_2F_1[5,5,10,w])(w\ {}_2F_1[1,1,2,w])^3 .
\end{aligned}
\tag{86}
$$

It would be interesting to determine the numeric coefficients in front of each term. The coefficient in front of the stress-tensor block to the $n$-th power appears to be equal to $(-1)^{n-1}2^{n-1}\frac{1}{n}$.

Naturally, the (holomorphic part of) the correlator has an expansion in powers of $\mu$ as well:

$$
e^{\Delta f(z)} = 1 + (\Delta f_1) + \left(\frac{1}{2}(\Delta f_1)^2 + \Delta f_2\right) + \dots .
\tag{87}
$$

The first and the second terms in this expansion are contributions of the identity and the stress tensor global blocks. The third term contains contributions from all double-trace operators composed out of the stress-tensor operator, which schematically have the form $T(z)\partial \dots \partial\ T(z) \equiv T\partial^s T$. This is because the OPE coefficients of two heavy operators $\mathcal{O}_H$ and these double-trace operators scale like $\Delta_H^2$; divided by the two-point function $\langle (T(z)\partial^s T(z))\ (T(0)\partial^s T(0)) \rangle \sim C_T^2$ this produces a factor of $\mu^2$. The sum over the infinite number of such operators with appropriate OPE coefficients, gives rise to the $\ln^2(1-w)$ terms in 85 which would not be obtained otherwise [72].

Let us briefly review how this works. The exact expressions for quasi-primaries at levels 4 (denoted by $\Lambda$) and level 6 (denoted by $\mathcal{O}_6^{(1,2)}$) can be found in appendix B of [71]:

$$
\Lambda = L_{-2}^2 - \frac{3}{5}L_{-4}, \qquad \mathcal{O}_6^{(1)} = -\frac{20}{63}L_{-6} - \frac{8}{9}L_{-4}L_{-2} + \frac{5}{9}L_{-3}L_{-3} .
\tag{88}
$$

Another quasi-primary at level 6 corresponds to a triple-trace operator. As expected, the normalization of these operators scales like $C_T^2$ at large $C_T$ and the OPE coefficient with a scalar operator of dimension $\Delta$ behaves like $a\Delta^2 + b\Delta$, where $a, b$ are some numbers. The appearance of the term proportional to $\Delta$ (as opposed to $\Delta^2$) is due to a piece which is linear in $T$, $\partial^s T(z)$. These operators, as well as double trace operators at higher levels, contribute to the $\mathcal{O}(\mu^2)$ terms in the expansion 87. Note that $\mathcal{O}(\Delta^2)$ term corresponds to $1/2(\Delta f_1)^2$, which comes from the exponentiation and is related to the phase shift at $\mathcal{O}(\mu)$. This exponentiation has been observed in [72], where it was pointed out that the large-spin behavior of the OPE coefficients of two operators $\mathcal{O}$ and a double trace operator of spin $s$ (discussed above) is known from the bootstrap:

$$
\lambda_{\mathcal{O}\mathcal{O}[T\partial^s T]}\lambda_{\mathcal{O}_H\mathcal{O}_H[T\partial^s T]} \simeq 2^{-2s}s^{-\frac{3}{2}}\left(\lambda_{\mathcal{O}\mathcal{O}T}\lambda_{\mathcal{O}_H\mathcal{O}_H T}\right)^2 \simeq \mu^2\, 2^{-2s}s^{-\frac{3}{2}}\Delta^2 .
\tag{89}
$$

At the same time, the small $z$ ($w \to 1$) behavior of conformal blocks in the T-channel contains a log term:

$$g_{\log} = \frac{\Gamma(2s)(1-w)^s}{\Gamma(s)^2} \ln(1-w) \approx \frac{2^{2s}\sqrt{s}}{2\sqrt{\pi}}(1-w)^s \ln(1-w), \tag{90}$$

where the approximation is valid at large $s$. Multiplying 89 by 90 and integrating over $s$ leads to a term of the form $\mu^2\Delta^2 \log^2(1-w)$, which corresponds to the $\mu^2\Delta^2 f_1^2$ term in 87 and comes from the exponentiation. Note that this exercise does not say anything about the $\mu^2\Delta f_2$ term; indeed, this term vanishes as $w \to 1$, as evident from the second line of 85.

Let us now consider the phase shift and see how the expansion in powers of $\mu$ works, term by term. In Section 2 we showed that the phase shift at $\mathcal{O}(\mu)$ is determined by the stress-tensor conformal block. We also observed above that the vacuum Virasoro block yields the expression to all orders in $\mu$ which is exactly the same as the one from the gravity calculation. Now we will reproduce the $\mathcal{O}(\mu^2)$ term in the phase shift from the $\mathcal{O}(\mu^2)$ correction to the $CFT_2$ correlator. This will be useful when dealing with higher dimensional CFTs. It is important to keep in mind the relation 48 between $x^\pm$ and $z, \bar{z}$, and also to include the conformal factor which arises when the cylinder is mapped to the plane.

Let us see how the pole can be expanded in powers of $\mu$, and integrated term by term:

$$\frac{1}{(x^+ - \Delta x^+)^\Delta} = \frac{1}{(x^+)^\Delta}\left[1 + \frac{\pi\Delta\mu}{x^+} + \left(\frac{\Delta(1+\Delta)\pi^2}{2(x^+)^2} + \frac{3\pi\Delta}{4x^+}\right)\mu^2 + \right.$$
$$\left. + \left(\frac{\Delta(1+\frac{3}{2}\Delta+\frac{1}{2}\Delta^2)\pi^3}{3(x^+)^3} + \frac{3\Delta(1+\Delta)\pi^2}{4(x^+)^2} + \frac{5\Delta\pi}{8(x^+)}\right)\mu^3 + \mathcal{O}(\mu^4)\right]. \tag{91}$$

After performing the Fourier transform, term by term, we arrive at

$$\int dx^+ \frac{e^{\frac{1}{2}ip^- x^+}}{(x^+ - \Delta x^+ - i\epsilon)^\Delta} = \frac{2^{1-\Delta}\pi}{\Gamma(\Delta)}\theta(p^-)e^{\frac{i}{2}\pi\Delta}(p^-)^{\Delta-1}$$
$$\times \left[1 + \frac{i\pi}{2}p^-\mu + \left(-\frac{\pi^2(p^-)^2}{8} + \frac{3\pi i p^-}{8}\right)\mu^2 + \dots\right]. \tag{92}$$

Note that all factors of $\Delta$ inside the square bracket disappear and we recover 81 expanded to $\mathcal{O}(\mu^2)$. This pattern continues to higher orders in $\mu$.

## 5 Phase shift and anomalous dimensions

For the case of a four-point correlator of two pairs of light operators $\mathcal{O}_1, \mathcal{O}_2$, the phase shift in the Regge limit can be related to the anomalous dimensions of the double twist operators - schematically $\mathcal{O}_1 \partial_{\mu_1} \dots \partial_{\mu_\ell} \partial^{2n} \mathcal{O}_2$ - exchanged in the S-channel expansion of the same correlator [57] (see also Appendix C of [52] for a review). In this Section we repeat the analysis in the case where one of the operators (say $\mathcal{O}_1 = \mathcal{O}_H$) is very heavy while the other operator $\mathcal{O}_2 = \mathcal{O}_L$ is light.

We claim that the correlator 44 admits the following impact parameter representation

$$\langle \mathcal{O}_H | \mathcal{O}_L(x_2)\mathcal{O}_L(x_3) | \mathcal{O}_H \rangle = \int_0^\infty d\xi \int_0^\xi d\bar{\xi} \, \mathcal{I}_{\xi,\bar{\xi}} \, e^{i\delta(\xi,\bar{\xi})}, \tag{93}$$

where

$$\mathcal{I}_{\xi,\bar{\xi}}(z,\bar{z}) = 4C(\Delta_L)\int_{M^+} \frac{d^d p}{(2\pi)^d}(-p^2)^{\Delta_L - \frac{d}{2}}e^{ip\cdot x}\xi\bar{\xi}(\xi^2 - \bar{\xi}^2)\delta\left(p.\bar{e} + \xi^2 + \bar{\xi}^2\right)\delta\left(\frac{p^2}{4} + \xi^2\bar{\xi}^2\right), \tag{94}$$

where $C(\Delta_L)$ is defined in 51 and $\delta(\xi, \bar{\xi})$ is the phase shift[8].

For simplicity, we consider the $d = 2$ case below, but the final formula is valid to leading order in $\mu$ in any number of dimensions. Let us consider the leading order $\mathcal{O}(\mu^0)$ term in 93, which corresponds to setting $\delta(\xi, \bar{\xi}) = 0$. This term should reproduce the contribution of the identity operator in the T-channel. One can use the identity

$$\delta\left(p.\bar{e} + \xi^2 + \bar{\xi}^2\right)\delta\left(\frac{p^2}{4} + \xi^2\bar{\xi}^2\right) = \frac{1}{|\xi^2 - \bar{\xi}^2|}\left[\delta\left(\frac{p^+}{2} - \xi^2\right)\delta\left(\frac{p^-}{2} - \bar{\xi}^2\right) + (p^+ \leftrightarrow p^-)\right] \tag{95}$$

to perform the integration over $p^+, p^-$. Substituting the result back into 93, using 48 and 94, and setting $d = 2$, leads to

$$\int_0^\infty d\xi \int d\bar{\xi}\, \mathcal{I}_{\xi,\bar{\xi}}(z, \bar{z}) = \frac{4}{\Gamma(\Delta_L)^2}\int_0^\infty d\xi \int_0^\xi d\bar{\xi}\,(\xi\bar{\xi})^{2\Delta_L - 1} z^{-\xi^2}\bar{z}^{-\bar{\xi}^2} + (z \leftrightarrow \bar{z}). \tag{96}$$

At the same time, the contribution of the identity operator in the cross channel can be computed directly using the known expressions for the cross-channel conformal blocks

$$g^{\Delta_{\mathcal{O}_H\mathcal{O}}, -\Delta_{\mathcal{O}_H\mathcal{O}}}_{\Delta = \Delta_{n,s}, J = \ell} = (z\bar{z})^{-\frac{\Delta_{n,s} - \ell}{2}}\left[z_2^\ell F_1(\Delta_L + n + \ell, \Delta_L + n + \ell, \Delta_H + \Delta_L + 2n + 2\ell, 1/z)\times\right.$$

$$\left. {}_2F_1(\Delta_L + n, \Delta_L + n, \Delta_H + \Delta_L + 2n, \bar{z}) + (z \leftrightarrow \bar{z})\right], \tag{97}$$

and the generalised free field theory OPE coefficients (eq. 43 in [70]). In the limit $\Delta_H \to \infty$, the blocks take a very simple form

$$g^{\Delta_{\mathcal{O}_H\mathcal{O}}, -\Delta_{\mathcal{O}_H\mathcal{O}}}_{\Delta = \Delta_{n,s}, J = \ell} \approx z^{-\frac{\Delta_H + \Delta_L + 2n + 2\ell}{2}}\bar{z}^{-\frac{\Delta_H + \Delta_L + 2n}{2}} + (z \leftrightarrow \bar{z}), \tag{98}$$

and the same is true about the OPE coefficients:

$$\lambda_{n,\ell}^2 \approx \frac{n^{\Delta_L - 1}(n + \ell)^{\Delta_L - 1}}{\Gamma(\Delta_L)^2}. \tag{99}$$

Hence, the contribution of the identity operator in the correlator can be written as the following sum in the S-channel

$$\langle\mathcal{O}_H|\mathcal{O}_L(x_3)\mathcal{O}_L(x_2)|\mathcal{O}_H\rangle = \frac{1}{\Gamma(\Delta_L)^2}\sum_{n,\ell \gg 1} n^{\Delta_L - 1}(n + \ell)^{\Delta_L - 1}z^{-n-\ell}\bar{z}^{-n} + (z \leftrightarrow \bar{z}) + \mathcal{O}(\mu). \tag{100}$$

In the Regge limit, where both $n, \ell \gg 1$, the sum can be replaced by an integral, allowing us to express the right hand side of 100 as

$$\frac{1}{\Gamma(\Delta_L)^2}\sum_{n,\ell \gg 1} n^{\Delta_L - 1}(n + \ell)^{\Delta_L - 1}z^{-n-\ell}\bar{z}^{-n} + (z \leftrightarrow \bar{z})$$

$$= \frac{1}{\Gamma(\Delta_L)^2}\int_0^\infty d\xi \int_0^\xi d\bar{\xi}\, n^{\Delta_L - 1}(n + \ell)^{\Delta_L - 1}z^{-n-\ell}\bar{z}^{-n}\frac{\partial n}{\partial\xi}\frac{\partial\ell}{\partial\bar{\xi}} + (z \leftrightarrow \bar{z}). \tag{101}$$

Note that the two expressions, 96 and 101 are identical, provided we perform the identification

$$\xi^2 = n + \ell, \qquad \bar{\xi}^2 = n. \tag{102}$$

---

[8]This is a straightforward generalization of the corresponding impact parameter representation introduced in [57]. Here we use $(\xi, \bar{\xi})$ in place of their $(h, \bar{h})$ for reasons which will become clear momentarily.

Moreover, the integral

$$\int_0^\infty d\xi d\bar{\xi} \, \mathcal{I}_{\xi,\bar{\xi}} \simeq \frac{1}{(-x^2)^{\Delta_L}} \tag{103}$$

reproduces the expected answer for the generalized free theory in the Regge limit.

We can now use the impact parameter representation 94 to write 93 in the form

$$\langle \mathcal{O}_H | \mathcal{O}_L(x_3) \mathcal{O}_L(x_2) | \mathcal{O}_H \rangle = \int_{M^+} \frac{d^d p}{(2\pi)^d} (-p^2)^{\Delta_L - \frac{d}{2}} e^{ipx} e^{i\delta(\xi(p),\bar{\xi}(p))}. \tag{104}$$

This expression represents the Fourier transform of 49, as long as we identify $\delta(\xi, \bar{\xi})$ with the phase shift which appears in 93. In 104 $\xi$ and $\bar{\xi}$ are related to $p$ via

$$-\frac{p^2}{4} = \xi^2 \bar{\xi}^2, \qquad \frac{p^+ + p^-}{2\sqrt{-p^2}} = \frac{1}{2}\left(\frac{\xi}{\bar{\xi}} + \frac{\bar{\xi}}{\xi}\right). \tag{105}$$

The second identity in 105 can also be written as

$$\frac{\bar{\xi}}{\xi} = e^{-L}, \tag{106}$$

where we used 68. Another way to rewrite 105 is

$$p^- = n, \qquad p^+ = n + \ell. \tag{107}$$

In the dual language (see [56,57,73]) this is simply saying that the $\mathcal{O}(\mu)$ correction to the energy of the bound state of a particle with momentum $\ell$ and radial excitation $n$ is $\delta$. Let us see why. The $\mathcal{O}(\mu)$ correction to the free result 100 comes from the anomalous dimensions of the double trace operators:

$$\langle \mathcal{O}_H | \mathcal{O}_L(x_3) \mathcal{O}_L(x_2) | \mathcal{O}_H \rangle =$$
$$= \frac{1}{\Gamma(\Delta_L)^2} \int_0^\infty d\xi d\bar{\xi} \, n^{\Delta_L - 1}(n+\ell)^{\Delta_L - 1} z^{-n-\ell} \bar{z}^{-n} \frac{\partial n}{\partial \xi} \frac{\partial \ell}{\partial \bar{\xi}} \left[1 - i\pi\gamma(\xi, \bar{\xi}) + \dots\right]. \tag{108}$$

The appearance of $-i\pi\gamma(\xi, \bar{\xi})$ in the brackets is due to the analytic continuation ($z \to e^{2\pi i}z$). From the discussion above it is clear that this results in

$$\langle \mathcal{O}_H | \mathcal{O}_L(x_3) \mathcal{O}_L(x_2) | \mathcal{O}_H \rangle = \int \frac{d^d p}{(2\pi)^2} (-p^2)^{\Delta - \frac{d}{2}} e^{ipx} \left[1 - i\pi\gamma(\xi(p), \bar{\xi}(p)) + \dots\right]. \tag{109}$$

Hence, to leading order in $\mu$, the anomalous dimension and the phase shift are related

$$\gamma_1(n, \ell) = -\frac{\delta_1(p)}{\pi}, \tag{110}$$

where the parameters are related by 107.

Let us now verify this relation for the phase shift computed in Section 2. As explained in [64], in two spacetime dimensions one can find the anomalous dimensions of the double trace operators $\mathcal{O}_H \partial_{\mu_1} \dots \partial_{\mu_\ell} \partial^{2n} \mathcal{O}_L$ by studying the eigenfunctions of the Hamiltonian in the $AdS_3$ spacetime with a conical defect 34. The result is

$$\gamma(n, \ell) = (\Delta_L + 2n)\left(\sqrt{1-\mu} - 1\right). \tag{111}$$

Note that in the Regge limit ($n \gg \Delta_L$) the $\mathcal{O}(\mu)$ term in the anomalous dimension becomes

$$\gamma_1 \approx -\mu n. \tag{112}$$

Using the identification 105 and the result for the phase shift 39 we recover 110.

Consider now $d = 4$. In this case, the leading behavior of the phase shift 73 (or, equivalently, 26) is given by

$$\delta_1 \simeq \mu \sqrt{-p^2} \frac{e^{-2L}}{\sinh L} = \mu \frac{n^2}{\ell}. \tag{113}$$

At the same time, the lightcone limit of the anomalous dimensions can be obtained from a gravity calculation [64, 67]. In [64, 67] there was a subtlety which involved the decomposition of the $\mathcal{O}_1 \partial_{\mu_1} \ldots \partial_{\mu_\ell} \partial^{2n} \mathcal{O}_2$ primary operator (with $\mathcal{O}_1, \mathcal{O}_2$ being light) into descendants. The result there was the dominance of the descendants of the type $\partial_{\mu_1} \ldots \partial_{\mu_{\ell/2}} \mathcal{O}_{1,2}$. When one of the operators is heavy, the sum is dominated by descendants of the type $\partial_{\mu_1} \ldots \partial_{\mu_\ell} \partial^{2n} \mathcal{O}_L$. The result is then

$$\gamma_1 \simeq \mu \frac{n^{\frac{d}{2}}}{\ell^{\frac{d-2}{2}}}, \tag{114}$$

which agrees with 112 in the $d = 2$ case and with 113 in the $d = 4$ case. (Note that while 114 was computed in the lightcone limit, $\ell \gg n \gg 1$, eq. 113 is valid in the Regge limit, $\ell \sim n \gg 1$).

In fact, it is very easy to see that in the lightcone limit 114 is equal to the phase shift in any $d$: it is sufficient to take the lightcone limit of 29:

$$\delta_1 \simeq \mu \sqrt{p^+ p^-} \, e^{-(d-1)L} \approx \mu \frac{n^{\frac{d}{2}}}{\ell^{\frac{d-2}{2}}}, \tag{115}$$

which is the same as 114.

This should be contrasted with the familiar story where the $\mathcal{O}(1/C_T)$ anomalous dimensions of the $\mathcal{O}_1 \partial_{\mu_1} \ldots \partial_{\mu_\ell} \partial^{2n} \mathcal{O}_2$ operators (with $\mathcal{O}_1, \mathcal{O}_2$ being light) are related to the phase shift observed in the scattering of *two* highly energetic particles [57]. The phase shift in this case is given by

$$\delta \simeq \frac{1}{C_T} \sqrt{p^2 \bar{p}^2} \, \Pi_{d-1; d-1}(L). \tag{116}$$

In the Regge limit the identification for the light operators is (we use the superscript "LL" below)

$$h^{LL} = n + \ell, \qquad \bar{h}^{LL} = n \tag{117}$$

and

$$16(h^{LL})^2 (\bar{h}^{LL})^2 = p^2 \bar{p}^2, \qquad e^{-L} = \frac{\bar{h}^{LL}}{h^{LL}}. \tag{118}$$

In $d = 4$ this recovers

$$\delta_1^{LL} \simeq \frac{n^4}{\ell(\ell + 2n)}, \tag{119}$$

which can also be obtained using conformal bootstrap [68, 69] (see also [34–40]).

Finally, let us observe that in $d = 2$, the anomalous dimension in 111 is not the same as the phase shift in 37 at second order and higher. In fact, in the next section, we will perform a second-order calculation of the anomalous dimension to show that this is also the case for general $d \geq 2$.

# 6 Anomalous dimensions of heavy-light double trace operators from gravity

The objective of this section is to obtain the anomalous dimensions of double-twist operators, schematically denoted by $[O_H O_L]_{n,\ell}$, with conformal dimensions $\Delta_{n,\ell} = \Delta_H + \Delta_L + 2n + \ell + \gamma_{n,\ell}$. We will follow the approach of [64,67] where the anomalous dimensions of double-twist operators built out of light operators, $[O_L O_L]_{n,\ell}$. For the case of heavy-light double-twist operators, we will focus on the limit, $\Delta_H \gg \ell \gg n \gg 1$, *i.e.*, the analogue of the lightcone limit for a very heavy operator.

Consider a generic double-twist primary of the form [48,70]

$$[O_H O_L]_{n,\ell} = \sum_{\ell_1 + \ell_2 = n; n_1 + n_2 = n} s_{\ell_1, n_1, \ell_2, n_2} \left[ \partial_{\mu_1} \dots \partial_{\mu_{\ell_1}} (\partial^2)^{n_1} O_H \right] \left[ \partial_{\mu_2} \dots \partial_{\mu_{\ell_2}} (\partial^2)^{n_2} O_L \right]. \quad (120)$$

Notice that the dominant contribution in the sum, in the limit $\Delta_H \gg 1$, comes from $\ell_1 = n_1 = 0$ and $n_2 = n$, $\ell_2 = \ell$. In other words, the tensor-products of the descendants of the light operator with $O_H$ form a primary[9]. To see this, consider the case $n = n_1 = n_2 = 0$. The coefficients $s_{\ell_1, 0, \ell_2, 0} \equiv s_{\ell_1, \ell_2}$ in the heavy limit can be computed using the results of [48,70]:

$$s_{\ell_1, \ell_2} \sim e^{\Delta_H} \Delta_H^{-\Delta_H - \ell_1 + \frac{1}{2}}. \quad (121)$$

Clearly, the non-zero $\ell_1$ is power-law suppressed (in $1/\Delta_H$) compared to the $\ell_1 = 0$ term[10]. In short, since the primary double-twist operator is $[O_H O_L]_{n,\ell}$ in the heavy limit is given by tensoring $O_H$ with a descendant $\partial_\mu \dots \partial_{\mu_\ell} (\partial^2)^n O_L$, we may compute the anomalous dimensions of such primaries by studying corrections to the energies of the descendants. In the gravity dual language, one should study corrections to the energy of the "descendants" of a free massive scalar field $\Phi$ in the $AdS_{d+1}$-Schwarzschild background. In this context, $\Phi$ is dual to the light operator $\mathcal{O}_L$ while the background is dual to the state created by the heavy operator $\mathcal{O}_H$.

To compute the corrections to the energy, we write the Hamiltonian in the form $H = H_0 + V$ with $H_0$ the Hamiltonian for a free massive scalar field in pure AdS and $V$ given by

$$V = \sum_{k \geq 1} \mu^k V^{(k)}, \quad (122)$$

where for $k = 1$

$$V^{(1)} = -\frac{1}{2} \int d^d x \, r \left[ \frac{1}{(r^2 + 1)^2} (\partial_t \Phi)^2 + (\partial_r \Phi)^2 \right], \quad (123)$$

while for $k \geq 2$

$$V^{(k)} = -\frac{k}{2} \int d^d x \, r^{d-1} \frac{1}{(r^2 + 1)^{k+1} r^{k(d-2)}} (\dot{\Phi})^2. \quad (124)$$

We refer the reader to Appendix C for more details on the derivation of these expressions. In the rest of the section we will employ standard perturbation theory techniques and obtain $\mathcal{O}(\mu)$ and $\mathcal{O}(\mu^2)$, *i.e.* the first and second order, corrections to the energy.

---

[9]Note that in the setup of [48,70], one needs to match the number of derivatives $\ell$ to the actual angular momentum of the dual primary double-twist operator. Here the situation is simpler as can be seen from this discussion.

[10]For general $n \neq 0$, it should be possible to prove a similar statement using the results of [48,70] but we shall not pursue it here.

## 6.1 First-order correction

Here we compute the $\mathcal{O}(\mu)$ correction to the anomalous dimensions in the lightcone limit, *i.e.*, $\ell \gg n \gg 1$. We thus focus on the $k = 1$ term of the potential. Note that the calculation in this case is similar to the one in [67] for double-twist operators build out of two light operators. In that case, [67] observed that $\gamma_{n,\ell}^{(1)} \sim n^{d/2}/\ell^{(d-2)/2}$. The $n$-dependence was deduced using numerical computations. Below we reproduce the respective result for heavy-light twist operators analytically (including the prefactor), by taking the large $\ell$ limit in a careful way in some intermediate step. This allows us to estimate the behavior of higher order terms, which is useful for the second-order calculation in the next section, where we compute $\gamma_{n,\ell}^{(2)}$ in the light-cone limit.

Consider the energy correction in first-order perturbation theory

$$\gamma_{n,\ell}^{(1)} = \langle n, \ell, j | V^{(1)} | n, \ell, j \rangle, \tag{125}$$

which in position space is given by

$$\gamma_{n,\ell}^{(1)} = -\frac{1}{2} \int dr \int d\Omega \, r \langle n, \ell, j | \left[ \frac{1}{(r^2+1)^2} (\partial_t \Phi)(\partial_t \Phi) + (\partial_r \Phi)(\partial_r \Phi) \right] | n, \ell, j \rangle. \tag{126}$$

From [64,67] we know that the leading $\ell$ behavior comes from the first term (i.e. $(\partial_t \Phi)(\partial_t \Phi)$ term).[11] Writing $\Phi$ in terms of the complete set of states $\psi_{n,\ell,j}$ (see Appendix C for more details), allows us to rewrite 126 as

$$\gamma_{n,\ell}^{(1)} = -\int dr \int d\Omega \, \frac{r}{(r^2+1)^2} (\partial_t \psi_{n,\ell,j}^*)(\partial_t \psi_{n,\ell,j}). \tag{127}$$

Performing the spherical integral - which gives unity - leads to

$$E_1 = -\frac{E_{n,\ell}^2}{N_{\Delta_L,n,\ell}^2} \int_0^\infty \frac{r^{1+2\ell}}{(r^2+1)^{2+\Delta_L+\ell}} \, {}_2F_1 \left[ -n, \Delta_L + \ell + n, \ell + \frac{d}{2}, \frac{r^2}{r^2+1} \right]^2, \tag{128}$$

where $E_{n,\ell}$ and $N_{\Delta_L,n,\ell}$ denote the energies and the normalisation coefficients of the unperturbed wavefunctions. Substituting their explicit expressions (see Appendix C) yields

$$E_1 = -\frac{(\Delta_L + 2n + \ell)^2 \Gamma(n + \ell + \frac{d}{2}) \Gamma(\Delta_L + n + \ell)}{n! \Gamma(\ell + (d/2))^2 \Gamma(\Delta_L + n - \frac{d-2}{2})} \times$$
$$\times \int_0^\infty \frac{r^{1+2\ell}}{(r^2+1)^{2+\Delta_L+\ell}} \, {}_2F_1 \left[ -n, \Delta_L + \ell + n, \ell + \frac{d}{2}, \frac{r^2}{r^2+1} \right]^2. \tag{129}$$

The prefactor in the lightcone limit $\ell \gg n \gg 1$ becomes

$$-\frac{(\Delta_L + 2n + \ell)^2 \Gamma(n + \ell + \frac{d}{2}) \Gamma(\Delta_L + n + \ell)}{n! \Gamma(\ell + (d/2))^2 \Gamma(\Delta_L + n - \frac{d-2}{2})} \approx -\frac{e^{2n} \ell^{-\frac{d}{2} + \Delta_L + 2n + 2} n^{\frac{d}{2} - \Delta_L - 2n - 1}}{(2\pi)}. \tag{130}$$

Understanding the behavior of the integral

$$I_0 \equiv \int_0^\infty \frac{r^{1+2\ell}}{(r^2+1)^{2+\Delta_L+\ell}} \, {}_2F_1 \left[ -n, \Delta_L + \ell + n, \ell + \frac{d}{2}, \frac{r^2}{r^2+1} \right]^2, \tag{131}$$

---

[11]One can show that the $(\partial_r \Phi)^2$ term in the large $\ell$ limit scales like $\ell^{-d/2}$ which is subleading compared to the $(\partial_r \Phi)^2$ term which scales like $\ell^{1-d/2}$. This can be shown explicitly by taking the large $\ell$ limit of the hypergeometric function in $\psi_{n,\ell,j}$.

requires a careful analysis. Naively, one may simply expand the hypergeometric function in the integrand for large $\ell$ to obtain

$$_2F_1\left[-n, \Delta_L + \ell + n, \ell + \frac{d}{2}, \frac{r^2}{r^2+1}\right] \approx (r^2+1)^{-n} + O(\ell^{-1}). \tag{132}$$

However, a careful look at the higher order terms of this expansion, shows that certain higher-order terms in the large $\ell$ expansion of $_2F_1$ contribute to the same order in $\ell$ after integration. These terms look like

$$(r^2+1)^{-n}\left(\frac{r^2}{\ell}\right)^a \tag{133}$$

for $a \geq 0$. Their contribution to the integral is

$$\int_0^\infty \frac{r^{1+2\ell}}{(r^2+1)^{2+\Delta_L+\ell}} (r^2+1)^{-2n}\left(\frac{r^2}{\ell}\right)^{a_1+a_2} =$$
$$= \ell^{-a_1-a_2} \frac{\Gamma(a_1+a_2+\ell+1)\Gamma(-a_1-a_2+2n+\Delta_L+1)}{2\Gamma(\ell+2n+\Delta_L+2)}, \tag{134}$$

which for large $\ell$ becomes

$$\frac{1}{2}\ell^{-\Delta_L-2n-1}\Gamma(-a_1-a_2+2n+\Delta_L+1). \tag{135}$$

Notice that the power of $\ell$ is independent of $a_1$ and $a_2$.

If we were only interested in the $\ell$ dependence, at this point we would conclude that at large $\ell$

$$E_1 \sim \frac{1}{\ell^{\frac{d-2}{2}}}. \tag{136}$$

To compute the $n$-dependence however, we need to keep track of the coefficients of all the terms in Eq. 133. In practice, we need the following approximate expression

$$_2F_1\left[-n, \Delta_\phi + \ell + n, \ell + \frac{d}{2}, \frac{r^2}{r^2+1}\right]$$
$$\approx (r^2+1)^{-n} \sum_{s=0}^n \frac{\Gamma(n+1)\Gamma\left(\frac{d}{2}+s-n-\Delta\right)}{\Gamma(s+1)\Gamma(-s+n+1)\Gamma\left(\frac{1}{2}(d-2(n+\Delta))\right)}\left(\frac{r^2}{\ell}\right)^s. \tag{137}$$

Substituting 137 in 131 leads to

$$I_0 \simeq \sum_{s_1,s_2=0}^n \frac{\Gamma(n+1)\Gamma\left(\frac{d}{2}+s_1-n-\Delta\right)}{\Gamma(s_1+1)\Gamma(-s_1+n+1)\Gamma\left(\frac{1}{2}(d-2(n+\Delta))\right)}$$
$$\times \frac{\Gamma(n+1)\Gamma\left(\frac{d}{2}+s_2-n-\Delta\right)}{\Gamma(s_2+1)\Gamma(-s_2+n+1)\Gamma\left(\frac{1}{2}(d-2(n+\Delta))\right)} \times \int_0^\infty \frac{r^{1+2\ell}}{(r^2+1)^{2+\Delta_L+\ell+2n}}\left(\frac{r^2}{\ell}\right)^{s_1+s_2}$$
$$\approx \ell^{-\Delta_L-2n-1} \frac{\Gamma(n+1)^2}{2\Gamma\left(\frac{1}{2}(d-2n-2\Delta)\right)^2} \times$$
$$\times \sum_{s_1,s_2=0}^n \frac{\Gamma\left(\frac{d}{2}+s_1-n-\Delta\right)\Gamma\left(\frac{d}{2}+s_2-n-\Delta\right)\Gamma(2n-s_1-s_2+\Delta_L+1)}{\Gamma(s_1+1)\Gamma(s_2+1)\Gamma(-s_1+n+1)\Gamma(-s_2+n+1)}, \tag{138}$$

where we first preformed the integral and then took the large $\ell$ limit. Now, let us sum over the $s_1$, to obtain

$$
I_0 \approx (-1)^{n+1} \frac{\pi}{2} \ell^{-\Delta_L - 2n - 1} \frac{\Gamma(n+1)}{\Gamma\left(\frac{1}{2}(d - 2(n + \Delta_L))\right)} \times
$$
$$
\times \sum_{s_2=0}^{n} \frac{\Gamma\left(\frac{d}{2} + n - s_2 + 1\right) \csc\left(\pi(\Delta_L + 2n - s_2)\right) \Gamma\left(\frac{d}{2} - n + s_2 - \Delta_L\right)}{\Gamma(s_2 + 1)\Gamma\left(\frac{d}{2} - s_2 + 1\right)\Gamma(n - s_2 + 1)\Gamma(-n + s_2 - \Delta_L)}, \tag{139}
$$

which can be shown to be equal to

$$
I_0 = \frac{1}{2} \ell^{-\Delta_L - 2n - 1} \frac{\Gamma\left(\frac{d}{2} + n + 1\right)\Gamma(n + \Delta_L + 1)}{\Gamma\left(\frac{d}{2} + 1\right)}
$$
$$
\times {}_3F_2\left(-\frac{d}{2}, -n, \frac{d}{2} - n - \Delta_L; -\frac{d}{2} - n, -n - \Delta_L; 1\right). \tag{140}
$$

The large $n$ limit of the ${}_3F_2$ is

$$
{}_3F_2\left(-\frac{d}{2}, -n, \frac{d}{2} - n - \Delta_L; -\frac{d}{2} - n, -n - \Delta_L; 1\right) \approx \frac{2n^{-\frac{d}{2}}\Gamma(d)}{\Gamma\left(\frac{d}{2}\right)} \tag{141}
$$

and so

$$
I_0 \simeq \frac{4\pi e^{-2n} \Gamma(d) n^{\Delta_L + 2n + 1}}{d\,\Gamma\left(\frac{d}{2}\right)^2}. \tag{142}
$$

Combining with the prefactor results in

$$
\gamma_{n,\ell}^{(1)} \simeq -\left(\frac{\Gamma(d)}{\Gamma(\frac{d}{2} + 1)\Gamma\left(\frac{d}{2}\right)}\right) \frac{n^{\frac{d}{2}}}{\ell^{\frac{d-2}{2}}}, \tag{143}
$$

which as expected agrees with the expression for the anomalous dimensions of the light-light twist-two operators in the lightcone limit given in [67].

## 6.2 Second-order correction

To second-order in $\mu$, there are two types of contributions to the energy:

$$
\gamma_{n,\ell}^{(2)} = \langle n, \ell | V^{(2)} | n, \ell \rangle + \sum_{E_{n_1,\ell_1,j_1} \neq E_{n,\ell,j}} \frac{|\langle n_1, \ell_1 | V^{(1)} | n, \ell \rangle|^2}{E_{n,\ell} - E_{n_1,\ell_1}}, \tag{144}
$$

i.e. there is a *first*-order-type correction from $V^{(2)}$ (since the coefficient is $\mu^2$) and there is a *second* order correction from $V^{(1)}$. In Appendix C, we show that the first term is subleading in $\ell$ for large $\ell$ to any order (not just quadratic in $\mu$). To be precise, we show that the $k$th-order contribution to the energy from $V^{(k)}$ behaves like $1/\ell^{(dk-2)/2}$ and is always subleading compared to the $k$th-order contribution from $V^{(1)}$ which behaves like $1/\ell^{(d-2)k/2}$. With that in mind, we will drop the first term in Eq. 144 and focus on

$$
\gamma_{n,\ell}^{(2)} \simeq \sum_{E_{n_1,\ell_1,j_1} \neq E_{n,\ell,j}} \frac{|\langle n_1, \ell_1 | V^{(1)} | n, \ell \rangle|^2}{E_{n,\ell} - E_{n_1,\ell_1}}. \tag{145}
$$

Similarly to the first order calculation, the leading large $\ell$ contribution comes from the $(\partial_t \Phi)^2$ term in $H_1$. We thus need the matrix element:

$$\langle n_1, \ell_1, j_1 | V^{(1)} | n, \ell, j \rangle \simeq - \int dr \int d\Omega \, \frac{r}{(r^2+1)^2} (\partial_t \psi_{n,\ell,j})(\partial_t \psi^*_{n_1,\ell_1,j_1}), \tag{146}$$

which due to the spherical integral picks up the Kronecker-$\delta$ contributions, $(\delta_{\ell,\ell_1} \delta_{j,j_1})$. As a result we can write:

$$\gamma^{(2)}_{n,\ell} \simeq_{\ell \gg 1} \sum_{n_1 \neq n} \frac{|\langle n_1, \ell | V^{(1)} | n, \ell \rangle|^2}{2(n - n_1)}. \tag{147}$$

Let us evaluate the matrix element in the numerator:

$$\langle n_1, \ell | V^{(1)} | n, \ell \rangle$$
$$= -e^{-i(E_n - E_{n_1})t} \frac{E_n E^*_{n_1}}{N_{\Delta_L,n} N_{\Delta_L,n_1}} \times \int_0^\infty dr \left\{ \frac{r^{2\ell+1}}{(r^2+1)^{2+\Delta_L+\ell}} \times \right.$$
$$\left. {}_2F_1 \left[ -n, \Delta_L + \ell + n, \ell + \frac{d}{2}, \frac{r^2}{r^2+1} \right] {}_2F_1 \left[ -n_1, \Delta_L + \ell + n_1, \ell + \frac{d}{2}, \frac{r^2}{r^2+1} \right] \right\}. \tag{148}$$

Similarly to the first order case, the large $\ell$ dependence can be found by considering the large $\ell$ limit of the hypergeometric function, which is equal to $(r^2+1)^{-n}$. Evaluating the integral using the large $\ell$ behavior of the hypergeometric function and taking again the large $\ell$ limit after performing the integration, gives:

$$\int_0^\infty dr \, \frac{r^{2\ell+1}}{(r^2+1)^{2+\Delta_L+\ell}} ({}_2F_1)({}_2F_1) \sim \ell^{-\Delta_L - n - n_1 - 1}. \tag{149}$$

We should of course consider the $\ell$-dependence from the normalization prefactors as well:

$$\frac{E_n E^*_{n_1}}{N_{\Delta_L,n} N_{\Delta_L,n_1}} \simeq \ell^{-(d/2)+\Delta_L + n + n_1 + 2}$$

$$\times \frac{(-1)^{n_1+n}}{\sqrt{\Gamma(n+1)\Gamma(n_1+1)\Gamma\left(-\frac{d}{2}+n+\Delta_L+1\right)\Gamma\left(-\frac{d}{2}+\Delta_L+n_1+1\right)}}. \tag{150}$$

Combining 150 and 149 we deduce that matrix element behaves in the large $\ell$ limit as follows

$$\langle n_1, \ell | V^{(1)} | n, \ell \rangle \sim \ell^{-\frac{d-2}{2}}, \tag{151}$$

which leads to the large $\ell$ dependence of the anomalous dimensions, *i.e.*,

$$\gamma^{(2)}_{n,\ell} \sim \frac{1}{\ell^{2 \times \frac{d-2}{2}}}. \tag{152}$$

Before moving on to discuss the $n$-dependence of $\gamma^{(2)}_{n,\ell}$, let us make a side comment on higher order contributions. It is easy to estimate, given the calculations above, the large $\ell$ dependence of the $k$-order term contribution from $V^{(1)}$. Assuming that in the large $\ell$-limit

$$\gamma^{(k)}_{n,\ell} \sim \langle n, \ell | V^{(1)} | n_1, \ell \rangle \langle n_1, \ell | V^{(1)} | n_2, \ell \rangle \ldots \langle n_k, \ell | V^{(1)} | n, \ell \rangle, \tag{153}$$

leads to

$$\gamma^{(k)}_{n,\ell} \sim \frac{1}{\ell^{k\frac{d-2}{2}}}. \tag{154}$$

It is plausible, based on the computation of the second order energy correction, that this is indeed the full leading contribution at order $k$. We have not pursued a rigorous general argument to order $k \geq 3$, but it is likely to be correct[12] .

Back to second order computations. Now that we have obtained the large $\ell$-dependence, we focus on finding the large $n$ dependence. To do so, we need to compute the matrix element in more detail,

$$
\langle n_1, \ell | V^{(1)} | n, \ell \rangle
$$

$$
\simeq -\frac{1}{2\ell^{\frac{d-2}{2}}} e^{-i(E_n - E_{n_1})t} \frac{(-1)^{n+n_1}}{\sqrt{n! \Gamma\left(-\frac{d}{2} + n + \Delta_L + 1\right)} \sqrt{n_1! \Gamma\left(-\frac{d}{2} + n_1 + \Delta_L + 1\right)}} \times \tag{155}
$$

$$
\sum_{s_1=0}^{n_1} \sum_{s=0}^{n} \frac{\Gamma(n-s+n_1-s_1+\Delta_L+1)\Gamma(n+1)\Gamma\left(\frac{d}{2}+s-n-\Delta\right)\Gamma(n_1+1)\Gamma\left(\frac{d}{2}+s_1-n_1-\Delta\right)}{\Gamma(s+1)\Gamma(-s+n+1)\Gamma\left(\frac{1}{2}(d-2(n+\Delta))\right)\Gamma(s_1+1)\Gamma(-s_1+n_1+1)\Gamma\left(\frac{1}{2}(d-2(n_1+\Delta))\right)}.
$$

The sum in 155 is nothing but a hypergeometric function, *i.e.*,

$$
\sum_{s_1=0}^{n_1} \sum_{s=0}^{n} \frac{\Gamma(n-s+n_1-s_1+\Delta_L+1)\Gamma(n+1)\Gamma\left(\frac{d}{2}+s-n-\Delta\right)\Gamma(n_1+1)\Gamma\left(\frac{d}{2}+s_1-n_1-\Delta\right)}{\Gamma(s+1)\Gamma(-s+n+1)\Gamma\left(\frac{1}{2}(d-2(n+\Delta))\right)\Gamma(s_1+1)\Gamma(-s_1+n_1+1)\Gamma\left(\frac{1}{2}(d-2(n_1+\Delta))\right)}
$$

$$
= \frac{\Gamma\left(\frac{d}{2}+n+1\right)\Gamma(n+\Delta_L+1)}{\Gamma\left(\frac{d}{2}+n-n_1+1\right)} \times {}_3F_2\left(-n, -\frac{d}{2}-n+n_1, \frac{d}{2}-n-\Delta_L; -\frac{d}{2}-n, -n-\Delta_L; 1\right). \tag{156}
$$

For even $d$, the $1/\Gamma\left(\frac{d}{2}+n-n_1+1\right)$ factor in the double sum, implies that the only non-zero terms are those for which $n_1 - n \geq -(d/2)$. The same conclusion can be reached by considering the ${}_3F_2$ hypergeometric; it is non-zero only when $n_1 - n \leq d/2$. Hence,

$$
\gamma_{n,\ell}^{(2)} \simeq
$$

$$
\simeq \frac{1}{8\ell^{d-2}} \sum_{-d/2 \leq n_1 \leq d/2} \frac{\Gamma\left(\frac{d}{2}+n+1\right)^2 \Gamma(n+\Delta_L+1)^2}{n! n_1! \Gamma\left(-\frac{d}{2}+n+\Delta_L+1\right)\Gamma\left(-\frac{d}{2}+n_1+\Delta_L+1\right)\Gamma\left(\frac{d}{2}+n-n_1+1\right)^2}
$$

$$
\times \frac{1}{(n-n_1)} \left[ {}_3F_2\left(-n, -\frac{d}{2}-n+n_1, \frac{d}{2}-n-\Delta_L; -\frac{d}{2}-n, -n-\Delta_L; 1\right) \right]^2. \tag{157}
$$

Notice that for fixed even $d$, this can be computed exactly in $n$. The answer is

$$
d = 2 : \gamma_{n,\ell}^{(2)} = -\frac{1}{8}(\Delta_L + 2n),
$$

$$
d = 4 : \gamma_{n,\ell}^{(2)} = -\frac{1}{8\ell^2}(\Delta_L + 2n - 1)\left[34n^2 + +34n(\Delta_L - 1) + \Delta_L(4\Delta_L + 1)\right],
$$

$$
d = 6 : \gamma_{n,\ell}^{(2)} = -\frac{1}{8\ell^4}(\Delta_L + 2n - 2)\big[786n^4 + 1572(\Delta_L - 2)n^3 + (6\Delta_L(164\Delta_L - 591) + 4046)n^2
$$

$$
+ 2(\Delta_L - 2)(3\Delta_L(33\Delta_L - 67) + 451)n + (\Delta_L - 1)\Delta_L\left(9\Delta_L^2 + 2\right)\big], \tag{158}
$$

which for large $n$ gives

$$
d = 2 : \gamma_{n,\ell}^{(2)} = -\frac{1}{4}n,
$$

$$
d = 4 : \gamma_{n,\ell}^{(2)} = -\frac{17}{2}\frac{n^3}{\ell^2},
$$

$$
d = 6 : \gamma_{n,\ell}^{(2)} = -\frac{393}{2}\frac{n^5}{\ell^4}. \tag{159}
$$

---

[12]Observe that the anomalous dimensions would then have the same large $\ell$ scaling as the phase shift computed in gravity, which at order $k$ behaves like $\delta_k \sim \ell^{-k(d-2)/2}$.

For general $d$, we observe that the large $n$ behavior for fixed $a = n_1 - n + (d/2)$ is

$$\left[ {}_3F_2\left(-n, -\frac{d}{2} - n + n_1, \frac{d}{2} - n - \Delta_L; -\frac{d}{2} - n, -n - \Delta_L; 1\right)\right]^2$$
$$\approx \left(n^{d-a}\frac{d!}{a!}\right)^2\left[1 - \frac{(d-a)(a+\Delta+1)}{n} + O(n^{-2})\right], \tag{160}$$

which together with the prefactor (which we need to keep to first subleading order in $n$) yields

$$\gamma_{n,\ell}^{(2)} \simeq -\frac{n^{d-1}}{\ell^{d-2}}\frac{d\Gamma(d+1)^2}{8}\sum_{a=0}^{(d-2)/2}\frac{1}{\Gamma(a+1)^2\Gamma(-a+d+1)^2}$$
$$= -\frac{n^{d-1}}{\ell^{d-2}}\frac{d\Gamma(d+1)^2}{8}\left[\frac{\Gamma(2d+1)}{\Gamma(d+1)^4} - \frac{{}_3F_2\left(1, -\frac{d}{2}, -\frac{d}{2}; \frac{d}{2}+1, \frac{d}{2}+1; 1\right)}{\Gamma\left(\frac{d+2}{2}\right)^4}\right]. \tag{161}$$

It is interesting to note that anomalous dimensions and phase shift agree up to a numerical factor in the lightcone limit.

## 7  Discussion

In this paper we consider the phase shift which a highly energetic scalar probe particle acquires as it travels near an asymptotically anti de Sitter black hole. The result has an expansion in powers of the black hole mass $\mu$. All terms in this expansion can be computed analytically. The dual, CFT interpretation of the phase shift involves the Fourier transform of a four-point function with two heavy operators, describing the black hole, and two light scalar operators, describing the probe particle. The expansion parameter $\mu$ corresponds to the ratio between the conformal dimension of the heavy operator and the central charge of the CFT, $\mu \sim \Delta_H/C_T$. The $\mu^k$ term in the expansion of the phase shift is related to the exchange of operators made out of $k$ copies of the stress tensor (with derivatives added).

The leading, $\mathcal{O}(\mu)$ phase shift, can be computed in a $d$-dimensional CFT using conformal Regge theory: the only contribution comes from the stress tensor. Generally, double-trace operators made out of the light scalar operator, also contribute to the four point function. The phase shift in the limit of high energies is insensitive to these contributions. We show that the CFT result exactly matches the gravity result.

In the case of a two-dimensional CFT we have more control over the CFT computation. We show that the vacuum Virasoro block in a CFT with a large central charge completely reproduces the phase shift to all orders in $\mu$. This should be contrasted with previous discussions of the heavy-heavy-light-light Virasoro block [64,66] (and also [74], where the entanglement entropy was computed in a heavy state and [78] ). In our setting we do not need to take the additional large temperature limit to observe the thermalization of the heavy state (see [75] for a recent discussion of thermalization in CFT). For us, it is sufficient for the CFT to be holographic. Presumably this is related to the fact that we focus on an observable which is not sensitive to the double trace operators in the T-channel (while the full four-point function is necessarily sensitive to their contributions). We observe that at least for one such observable (the phase shift) the answer is universal and completely matches the one predicted by the dual gravity.

The two-dimensional case is quite instructive, because it explicitly shows that generally an infinite number of multiple-trace operators must be summed and then the result should be analytically continued. It also shows that the Virasoro vacuum block reproduces the gravity phase shift to all orders in $\mu$ (of course we have known that the double trace operators made

out of the light scalars do not contribute to the phase shift at leading order in $\mu$). It would be interesting to see if this remains true in higher dimensions.

The two-dimensional case also provides us with an example where the phase shift and anomalous dimensions of double trace operators differ beyond $\mathcal{O}(\mu)$ (we explain why they must be the same at $\mathcal{O}(\mu)$ in Section 5). Note that the functional behavior of the anomalous dimensions matches the one inferred from the phase shift, at least in the light cone limit where we computed anomalous dimensions to $\mathcal{O}(\mu^2)$. For example, in $d = 4$, $\gamma_2 \sim \mu^2 n^3 / \ell^2$, which is what one would infer from $\delta_2$ using 107. The numerical coefficients are different beyond $\mathcal{O}(\mu)$; it would be interesting to relate the anomalous dimensions and the phase shift directly uring the conformal bootstrap approach[13].

In our discussion of the two-dimensional CFT we observed that $\mathcal{O}(\mu^k)$ term in the correlator has a simple structure 86. Namely, the $\mathcal{O}(\mu^2)$ term is a combination of the product of spin-1 and spin-3 global blocks and the square of the spin-2 global block. The structure is similar at higher orders. Unfortunately we could not efficiently guess the structure of the correlator in higher dimensions. In fact, it is probably sensitive to the three-point function of the stress tensor. On the other hand, in holographic theories this three-point function is uniquely determined by unitarity. It would be interesting to directly sum over the multiple stress tensor contributions and see if the answer is universal (and reproduces the black hole result). We do need the corresponding OPE coefficients – perhaps the methods of [76, 77] will be helpful here.

More generally, the setup of this paper, where $\mu \sim \Delta_H / C_T$ is fixed in the limit of large central charge, identifies an interesting scaling limit in holography, where a subset of loop diagrams in the bulk survives. There are still Witten diagrams which are suppressed as we take the large central charge limit. The phase shift calculation computes one useful observable in this scaling limit. It would be interesting to find other observables of this type; it remains to be seen whether this can teach us something about quantum gravity in the bulk.

It would also be interesting to relate the results of this paper to various other developments. For example, in the setup of this paper one should be able to see how the phase shift ceases to be real, as the test particle falls into a black hole. This is presumably related to inelastic high-energy scattering studied e.g. in [23–30]. It would also be interesting to explore the relation of our work, where a heavy state exhibits features of a black hole, to the fuzzball proposal (see e.g. [80–82]).

## Acknowledgements

We benefitted from discussions with Monica Guica, Daniel Jafferis, Robin Karlsson, David Kutasov, Daliang Li, Emil Martinec, Joao Penedones, Petar Tadic, Junpu Wang, Sasha Zhiboedov. G. N. is supported by Simons Foundation Grant to HMI under the program "Targeted Grants to Institutes". The work of A.P. is supported in part by the Irish Research Council Laureate Award. This work was also supported in part by the National Science Foundation under Grant No. NSF PHY-1748958. M.K. and A.P. thank the Galileo Galilei Institute for Theoretical Physics, NORDITA, Simons Center For Geometry and Physics, KITP Santa Barbara for hospitality during the completion of this work.

---

[13]We emphasize that the large spin behavior here is different from the one suggested by the usual lightcone bootstrap, because we explicitly assume $\Delta_H \gg \ell$. For example, in $d = 4$, $\gamma_1 \sim \delta_1 \sim \mu/\ell$, while the standard behavior for $\ell \gg \Delta_{1,2}$ is $\gamma_1^{lightcone} \sim 1/\ell^2$. Presumably, there is a crossover scale, set by $\Delta_H$, beyond which the spin dependence of the double trace operators takes the usual form $\gamma \sim 1/\ell^{\tau_m}$, where $\tau_m$ is the minimal twist of contributing operators in the T-channel.

# A   Integrals: the bulk phase shift in gravity

We start by computing the quadratic $\mu$-term for the phase shift. As explained in the main text, we have :

$$\delta_2 = \mu^2 \frac{1}{2} \sqrt{-p^2} \left( 2c_2 \left.\frac{\partial \delta}{\partial v_0^2}\right|_{v_0^2=0} + c_1^2 \left.\frac{\partial^2 \delta}{\partial (v_0^2)^2}\right|_{v_0^2=0} \right) =$$

$$= -\mu^2 \frac{1}{2} \sqrt{-p^2}\, b^{-2d+5} \times$$

$$\times \int_0^1 dy \left\{ \frac{(1-y^d)(b^2+y^d)}{\sqrt{1-y^2}(b^2+y^2)} - \frac{\sqrt{1-y^2}(b^2+y^d)}{(b^2+y^2)^3}(d(b^2+y^2)+2(y^d-y^2))+ \right.$$

$$\left. +\frac{(1-y^d)(2(d-2)(1-y^2)+(1-y^d))}{4(1-y^2)^{\frac{3}{2}}(b^2+y^2)} \right\} . \tag{162}$$

It is convenient to express the integrand, *i.e.*, the terms within the curly brackets, as:

$$\{\cdots\} = \frac{\partial Q(y)}{\partial y} - \frac{1}{4}(2d-3)(2d-1)\, y^{2(d-2)} \sqrt{1-y^2}(b^2+y^2)^{-1}, \tag{163}$$

where

$$Q(y) \equiv \frac{-y\sqrt{1-y^2}(1-y^d)}{4(y^2+b2)}\left[ 2d(b^2+y^d) - \frac{1-y^{d-2}}{1-y^2}(b^2+y^2) + 2(d-1)(y^2+b^2y^{d-2}) \right]. \tag{164}$$

It is easy to see that the total derivative term evaluates to zero and we are left with

$$\delta_2 = \mu^2 \sqrt{-p^2}\, b^{-2d+5} \frac{1}{8}(2d-3)(2d-1) \int_0^1 dy\, y^{2(d-2)} \sqrt{1-y^2}(b^2+y^2)^{-1} =$$

$$= \mu^2 \sqrt{-p^2}\, b^{-2d+3} \frac{1}{8}(2d-3)(2d-1)\, B\left[\frac{2d-3}{2}, \frac{3}{2}\right]\, {}_2F_1[1, \frac{2d-3}{2}, d, -\frac{1}{b^2}], \tag{165}$$

which is proportional to the propagator for a particle of mass-squared equal to $(2d-3)$ in a hyperbolic space of the same dimensionality. To see this, one needs to use once more 27 but now set $a_1 = 2d-3$, $a_2 = d-2$. The result is:

$$\delta_2 = \mu^2 \sqrt{-p^2}\, \frac{1}{8}(2d-3)(2d-1)\, B\left[\frac{2d-3}{2}, \frac{3}{2}\right]\, 2^{2d-3}\, e^{-(2d-3)L}\, {}_2F_1[2d-3, d-2, d, e^{-2L}]$$

$$\implies \qquad \delta_2 = \mu^2 \frac{(2d-3)(2d-1)}{4} \frac{\pi^{d-1}}{\Gamma[d-1]} \sqrt{-p^2}\Pi_{2d-3,2d-3}(L) . \tag{166}$$

Evaluating a few higher order terms in the $\mu$ expansion reveals a pattern which allows us to write:

$$\delta(\sqrt{-p^2}, L) = \sum_{k=0}^{\infty} \frac{\mu^k}{k!} \frac{2\Gamma\left[\frac{dk+1}{2}\right]}{\Gamma\left[\frac{k(d-2)+1}{2}\right]} \frac{\pi^{\frac{k(d-2)+2}{2}}}{\Gamma[\frac{k(d-2)+2}{2}]} \sqrt{-p^2}\, \Pi_{k(d-2)+1,k(d-2)+1}(L) =$$

$$= 2b\sqrt{-p^2} \int_0^1 dy\, \frac{\sqrt{1-y^2}}{y^2+b^2} \left\{ \sum_{k=0}^{\infty} \left( \frac{\mu^k}{k!} \frac{\Gamma[\frac{dk+1}{2}]}{\Gamma[\frac{dk+1}{2}-k]} b^{-k(d-2)} \right) y^{k(d-2)} \right\}, \tag{167}$$

where the integral expression follows from an integral representation of the hypergeometric functions ${}_2F_1$. We have explicitly checked that 167 leads to the correct result in a number of dimensions and orders.

# B  Closed form result for the phase shift in $d = 4$

For the sake of completeness, we add here the closed form expression for the bulk phase shift in $d = 4$ dimensions. The final result is first found in terms of Appell $F_1$ hypergeometric functions, and then expressed in terms of elliptic integrals.

*Method 1: Appell $F_1$ functions.* Let us start from the expression for the phase shift as given in 19 and substitute $d = 4$.

$$\delta = \sqrt{-p^2} \frac{2b\sqrt{1-v_0^2}}{v_0^2} \int_0^1 dy \frac{\sqrt{1-y^2}\sqrt{1-my^2}}{(y^2-y_1^2)(y^2-y_2^2)}, \tag{168}$$

where we set

$$\kappa = \frac{v_0^2}{1-v_0^2} = \left( \frac{b^2}{2\mu}(1+\sqrt{1-\frac{4\mu}{b^2}}) - 1 \right)^{-1}, \tag{169}$$

and defined $y_{1,2}$ are as solutions of the following algebraic equation:

$$y_{1,2}^4 - \frac{y_{1,2}^2}{v_0^2} - \frac{b^2}{\kappa} = 0,$$

$$y_{1,2}^2 = \frac{1 \pm \sqrt{1+4b^2v_0^2(1-v_0^2)}}{2v_0^2} = \frac{b^2}{4\mu}(1+\sqrt{1-\frac{4\mu}{b^2}})(1\pm\sqrt{1+4\mu}). \tag{170}$$

Using the method of "partial fractions"

$$\frac{1}{(y^2-y_1^2)(y^2-y_2^2)} = \frac{1}{y_1^2-y_2^2}\left( \frac{1}{y^2-y_1^2} - \frac{1}{y^2-y_2^2} \right), \tag{171}$$

we can split the integral into two integrals of the form:

$$I = \int_0^1 dy (1-y^2)^{\frac{1}{2}}(1-\kappa y^2)^{\frac{1}{2}}\left( 1 - \frac{y^2}{y_{1,2}^2} \right)^{-1}, \tag{172}$$

which are integral representations of the hypergeometric with two variables (AppellF1). Precisely we obtain:

$$\delta = \sqrt{-p^2}\frac{b\sqrt{1-v_0^2}}{v_0^2}\frac{\pi}{2}\frac{1}{y_1^2 y_2^2}\frac{\left( \frac{1}{y_1^2}F_1[\frac{1}{2};-\frac{1}{2},1;2;\kappa,\frac{1}{y_1^2}] - \frac{1}{y_2^2}F_1[\frac{1}{2};-\frac{1}{2},1;2;\kappa,\frac{1}{y_2^2}] \right)}{\frac{1}{y_1^2}-\frac{1}{y_2^2}}, \tag{173}$$

with $F_1$ the Appell $F_1$ function.

*Method 2: Elliptic integrals*

In this case, it is easier to separately compute the time delay and deflection. Starting from 9 and eliminating $\mu$ in favor of $v_0^2$ defined in 20, we can write the time delay as:

$$\Delta t = -\frac{2b^2\sqrt{1-v_0^2}}{v_0^2}\frac{\sqrt{1+b^2}}{b}\int_0^1 \frac{dy}{(y^2-y_1^2)(y^2-y_2^2)\sqrt{(1-y^2)(1-\kappa y^2)}}, \tag{174}$$

where $y_{1,2}$ are defined as solutions of the following algebraic equation:

$$y_{1,2}^4 - \frac{y_{1,2}^2}{v_0^2} + b^2(1-\frac{1}{v_0^2}) = 0,$$

$$y_{1,2}^2 = \frac{1 \pm \sqrt{1+4b^2v_0^2(1-v_0^2)}}{2v_0^2}, \tag{175}$$

where again

$$\kappa = \frac{v_0^2}{1-v_0^2}. \tag{176}$$

Using the fact that

$$\frac{1}{(y^2-y_1^2)(y^2-y_2^2)} = \frac{1}{y_1^2-y_2^2}\left(\frac{1}{y^2-y_1^2}-\frac{1}{y^2-y_2^2}\right), \tag{177}$$

it is easy to see that 174 can be expressed as the difference of two complete elliptic integrals of the third kind:

$$\Delta t = -\frac{\sqrt{1+b^2}}{b\sqrt{1-v_0^2}} \times \left\{ \left(1-\sqrt{1+4b^2 v_0^2(1-v_0^2)}\right)\Pi\left[\frac{2v_0^2}{1+\sqrt{1+4b^2 v_0^2(1-v_0^2)}}, \frac{v_0^2}{1-v_0^2}\right] - \right.$$
$$\left. (1+\sqrt{1+4b^2 v_0^2(1-v_0^2)})\Pi\left[\frac{2v_0^2}{1-\sqrt{1+4b^2 v_0^2(1-v_0^2)}}, \frac{v_0^2}{1-v_0^2}\right]\right\}, \tag{178}$$

with the complete elliptic integral of the third kind defined as:

$$\Pi[\tilde{\kappa},\kappa] = \int_0^1 \frac{dt}{(1-\tilde{\kappa}t^2)\sqrt{1-\kappa t^2}\sqrt{1-t^2}}. \tag{179}$$

Let us turn to the integral defining the deflection of the particle in four dimensions which can be expressed as:

$$\Delta\phi = \frac{2}{\sqrt{1-v_0^2}}\int_0^2 \frac{dy}{\sqrt{1-y^2}\sqrt{1-\kappa y^2}} = \frac{2}{\sqrt{1-v_0^2}}K[\kappa], \tag{180}$$

with $\kappa$ defined in 176 and $K[\kappa]$ the complete elliptic integral of the second kind.

## C  Details on anomalous dimension calculation in the bulk

### C.1  Perturbations of Hamiltonian

Let us begin with the scalar action:

$$I = \int d^{d+1}x\,\sqrt{-g}L = \int d^{d+1}x\,\sqrt{-g}\left[-\frac{1}{2}g^{\mu\nu}\partial_\mu\Phi\partial_\nu\Phi - \frac{1}{2}m^2\Phi^2\right] = \sum_{k=0}\mu^k\int d^{d+1}x\,r^{d-1}\,L_k. \tag{181}$$

Expanding the metric in Eq. 1 in powers of $\mu$ leads to

$$L = \sum_{k=0}\mu^k\int d^{d+1}x\,r^{d-1}\,L_k, \tag{182}$$

where $L_0$ is the Lagrangian of the scalar field in AdS

$$L_0 = \frac{1}{2}\frac{1}{r^2+1}(\partial_t\Phi)^2 - \frac{1}{2}(r^2+1)(\partial_r\Phi)^2 - \frac{1}{2r^2}\gamma^{ij}\partial_i\Phi\partial_j\Phi - \frac{1}{2}m^2\Phi^2, \tag{183}$$

while

$$L_1 = \frac{1}{2} \frac{1}{r^{d-2}} \left[ \frac{1}{(r^2+1)^2} (\partial_t \Phi)^2 + (\partial_r \Phi)^2 \right], \tag{184}$$

and for $k \geq 2$

$$L_k \equiv \frac{1}{2} \frac{1}{r^{(d-2)k}(r^2+1)^{k+1}} (\partial_t \Phi)^2. \tag{185}$$

Next, we compute the Hamiltonian to every order in $k$.

The scalar field's stress tensor is

$$T^{(\Phi)}_{\mu\nu} = \partial_\mu \Phi \partial_\nu \Phi + g_{\mu\nu} L \tag{186}$$

and so the conserved energy is

$$H = \int d^d x \ \sqrt{h} n^t T^{(\Phi)}_{tt} = \int d^d x \ r^{d-1} \left[ f^{-1} \dot{\Phi}^2 - L \right], \tag{187}$$

where $h$ is the induced metric while $n$ is the normal vector $n^\mu \partial_\mu = (1/\sqrt{-g_{tt}})\partial_t$. In canonical quantization, we define the conjugate momentum at a constant time slice by

$$\Pi_\phi \equiv \frac{\delta L}{\delta \dot{\Phi}} = f^{-1} \dot{\Phi}. \tag{188}$$

So the Hamiltonian can be expressed as

$$H = \int d^d x \ r^{d-1} \left[ \Pi_\Phi \dot{\Phi} - L \right]. \tag{189}$$

We now substitute $L = L_0 + \sum_{k=1}^{\infty} \mu^k L_k$ and rewrite the Hamiltonian in terms of the canonical momenta

$$H = \int d^d x \ r^{d-1} \left[ \frac{1}{2}(\Pi_\Phi)^2(r^2+1) + \frac{1}{2}(r^2+1)(\partial_r \Phi)^2 + \frac{1}{2}m^2 \Phi^2 + \frac{1}{2r^2}\gamma^{ij} \partial_i \Phi \partial_j \Phi \right.$$
$$\left. - \sum_{k=1}^{\infty} \mu^k L_k - \frac{1}{2}\mu^2 \frac{1}{(r^2+1)r^{2(d-2)}}(\Pi_\Phi)^2 \right]. \tag{190}$$

Observe that the first line is simply the Hamiltonian on pure AdS, which we denote as the unperturbed Hamiltonian, $H_0$. Hence, $H = H_0 + V$ where

$$V = -\int d^d x \ r^{d-1} \left[ \sum_{k=1}^{\infty} \mu^k L_k + \frac{1}{2}\mu^2 \frac{1}{(r^2+1)r^{2(d-2)}}(\Pi_\Phi)^2 \right]. \tag{191}$$

More explicitly,

$$V = \sum_{k \geq 1} \mu^k V^{(k)}, \tag{192}$$

where for $k = 1$

$$V^{(1)} = -\frac{1}{2} \int d^d x \ r \left[ \frac{1}{(r^2+1)^2} (\partial_t \Phi)^2 + (\partial_r \Phi)^2 \right], \tag{193}$$

while for $k \geq 2$

$$V^{(k)} = -\frac{k}{2} \int d^d x \ r^{d-1} \frac{1}{(r^2+1)^{k+1} r^{k(d-2)}} (\dot{\Phi})^2. \tag{194}$$

## C.2 Unperturbed states

The unperturbed wave functions are [49, 50];

$$\psi_{n,\ell,j}(t,r,\Omega) = N^{-1}_{\Delta_L,n,\ell} e^{-iE_{n,\ell}t} Y_{L,j}(\Omega) \frac{r^\ell}{(1+r^2)^{\frac{\Delta_L+\ell}{2}}} \,_2F_1\left[-n, \Delta_L+\ell+n, \ell+\frac{d}{2}, \frac{r^2}{r^2+1}\right], \quad (195)$$

where

$$E_{n,\ell} = \Delta_L + 2n + \ell, \quad (196)$$

and

$$N_{\Delta,n,\ell} = (-1)^n \left[\frac{n!\Gamma(\ell+(d/2))^2\Gamma(\Delta+n-\frac{d-2}{2})}{\Gamma(n+\ell+\frac{d}{2})\Gamma(\Delta+n+\ell)}\right]^{1/2}. \quad (197)$$

The eigenstates are defined as

$$|n,\ell,j\rangle \equiv a^\dagger_{n,\ell,j}|0\rangle, \quad (198)$$

with the position space representation

$$\psi_{n,\ell,j}(t,r,\Omega) = \langle x|n,\ell,j\rangle. \quad (199)$$

A general state in position space is then

$$\Phi = \sum_{n,\ell,j}\left[a^\dagger_{n,\ell,j}\psi_{n,\ell,j} + a_{n,\ell,j}\psi^*_{n,\ell,j}\right]. \quad (200)$$

We will also need to define the composite operator $\Phi(x)^2$ (and various versions of this operator with derivatives). Using the normal-ordered product, we may write

$$\Phi^2 \equiv \sum_{n,\ell,j;n',\ell',j'}\left[a^\dagger_{n,\ell,j}a^\dagger_{n',\ell',j'}\psi_{n,\ell,j}\psi_{n',\ell',j'} + a^\dagger_{n',\ell',j'}a_{n,\ell,j}\psi^*_{n,\ell,j}\psi_{n',\ell',j'}\right.$$
$$\left.+ a^\dagger_{n,\ell,j}a_{n',\ell',j'}\psi_{n,\ell,j}\psi^*_{n',\ell',j'} + a_{n,\ell,j}a_{n',\ell',j'}\psi^*_{n,\ell,j}\psi^*_{n',\ell',j'}\right]. \quad (201)$$

As a result we see for instance, that:

$$\langle n,\ell,j|(\partial_t\Phi)^2|n_1,\ell,j\rangle = 2(\partial_t\psi^*_{n_1,\ell,j})(\partial_t\psi_{n,\ell,j}). \quad (202)$$

## C.3 Order $\mu^k$ term from $V^{(k)}$

Let us focus on the $k \geq 2$ term coming purely from $V^{(k)}$. The contribution to the energy at order $k$ from this term is

$$\langle n,\ell,j|V^{(k)}|n,\ell,j\rangle = -\frac{k}{2}\frac{E^2_{n,\ell}}{N^2_{\Delta_\phi,n,\ell}}\int_0^\infty \frac{r^{d(1-j)+2k-1+2\ell}}{(r^2+1)^{k+1+\Delta_\phi+\ell}} \,_2F_1\left[-n, \Delta_\phi+\ell+n, \ell+\frac{d}{2}, \frac{r^2}{r^2+1}\right]^2. \quad (203)$$

As explained in Sec. 6.1, the large $\ell$ dependence of the hypergeometric function is proportional to $(r^2+1)^{-n}$. Performing the integration and taking the large $\ell$ limit leads to

$$\langle n,\ell,j|V^{(k)}|n,\ell,j\rangle \sim \frac{1}{\ell^{\frac{dk-2}{2}}} = \frac{1}{\ell^{\frac{k(d-2)}{2}}} \times \frac{1}{\ell^{k-1}}. \quad (204)$$

As discussed in Sec 6.2 around Eq. 153, the $V^{(1)}$ term at order $k$ behaves like $\ell^{-k\frac{d-2}{2}}$ at large $\ell$, hence the $\langle n,\ell,j|V^{(k)}|n,\ell,j\rangle$ yields a subdominant contribution in the large $\ell$ limit.

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
