# Peer review of "Black Holes, Heavy States, Phase Shift and Anomalous Dimensions"

_SciPost Physics, doi:SciPost Phys. 6, 065 (2019)_

## Round 2 · Referee Report · Anonymous (Referee 1) · 2019-5-5

Strengths

1.- This manuscript does non-trivial tests of the AdS/CFT correspondence. In particular they compare the motion of null geodesics on an AdS black hole to the shift that appears in the Regge limit of 4-point functions in the CFT.

2.- The authors present a detailed comparison on how this phase shift is related to the anomalous dimension of certain CFT operators.

Weaknesses

1.- Even for an expert on AdS/CFT, the paper is at times a bit difficult to follow. I would have preferred a more detailed introduction on the state of the art on comparing the geodesic motion to the phase shift in the Regge limit. The authors do give a complete list of reference, so the background material is clear; it is just that the current presentation makes it less accesible for those not working on this specific aspect of AdS/CFT.

2.- There are no figures, which makes the text and setup a bit more difficult to follow. The authors do explain in words clearly the technical setup, but sometimes a diagram that accompanies the text is extremely useful. For example, a figure for the shape of geodesics considered in sec 2 would be nice, and a figure at the start of section 3.1 that depicts the positions of the operators and the limits taken.

Report

I recommend this paper for publication. My questions and weakness are relatively minor, and I think do not impact the scientific results presented.

Requested changes

I just have some minor questions/remarks:

1. Section 2.3 focus on 3D gravity, and I have a few questions. First, all results are done only for μ<1. What are the expression for eqn (2.38) for BTZ (i.e. μ>1)? Could one compute the phase shift in close form?

2. How does (2.39) compare to (2.33)? Conceptually, the answer is not obvious to me since in d>2 the solution is a honest black hole, but in d=2 the results reported are only reported for a conical defect.

3. There is a typo in the first paragraph of section 2.3. I think the authors meant the to refer to equation (2.10) and not (2.3).

4. In relation to the weakness I mention above, I thought Section 3 was a bit more difficult to follow. It would be great if there is a figure depicting the kinematics of section 3.1.

5. I didn't understand the paragraph above eqn. (3.9). Why do the coordinates x0 and φ depend on μ? Up to there it seems like finding (3.9) has nothing to do with the value of μ. I suspect I'm misunderstanding the content of that paragraph.

6. In the last line of the discussion, the authors present some future directions. But some aspects were unclear. Could the authors elaborate on what is the potential relation of the work here with the fuzzball proposal?

---

## Round 2 · Referee Report · Anonymous (Referee 2) · 2019-5-14

Strengths

1. The question that the authors address is inherently valuable in the study of holography, specifically the incorporation of gravitational backreaction into scattering observables.

2. The authors nicely match results in CFT and gravity, giving further confidence in the outcome. Moreover they show facility with various of-the-moment techniques and the computations are impressive yet easy to follow.

3. The specialization of their endeavor to two-dimensions gives all-orders results which are intriguing, and suggest relations with other similar computations in two dimensions.

4. The logic of the computations is pretty clear, and the writing is unfussy.

Weaknesses

1. In d=2, the authors were not completely clear about whether their computations actually apply for black hole backgrounds (as opposed to conical defects), despite the title, and if they do, whether this follows from the usual naive analytic continuation or involves other subtleties.

2. I would have liked more insight into how their results for the phase shift, in ΔH/c1 perturbation theory, is expected to make contact with the expansion of other CFT observables. Also, it wasn't clear to me what role higher multi-traces would play at higher orders in perturbation theory, and thus more generally what resummation in d>2 would mean.

Report

I enjoyed this paper for the reasons given above, and also because I had not realized that this was an open question. So I think the authors identified a worthwhile regime in which to study the connection amongst phase shifts and anomalous dimensions of CFT multi-trace operators.

Requested changes

1. Below 6.27, Kronecker should be capitalized.

---

## Round 3 · Author Response

We would like to thank the referees for their comments. In response, we made the following changes such that some aspects of the paper are clearer.

---

## Round 3 · List of Changes

1. We added a paragraph at the end of Section 2.3 discussing the difference between the BTZ and the conical defect cases.

2. We added a sentence below eq. 2.39 emphasising its agreement with eq. 2.33.

3. We changed the first sentence of Section 2.3 to make it clearer.

4. We added Fig. 1 as requested by the referee.

5. We removed the first sentence below eq. 3.8 and modified the next sentence.

6. We modified the last sentence of Section 7 as requested.

7. We also capitalized “kronecker”, as requested by the referee.

Resubmission 1812.03120v3 on 16 May 2019
Submission 1812.03120v2 on 26 March 2019

---

## Editorial Decision

published